# Understanding Class Bias Amplification in Graph Representation Learning

**Shengzhong Zhang**                                           *szzhang@nuaa.edu.cn*
*College of Computer Science and Technology*
*Nanjing University of Aeronautics and Astronautics*

**Wenjie Yang**[*]                                            *yangwj24@m.fudan.edu.cn*
*School of Data Science, Fudan University*

**Yimin Zhang**                                               *yiminzhang20@fudan.edu.cn*
*School of Data Science, Fudan University*

**Hongwei Zhang**                                             *zhanghw.hongwei@gmail.com*
*School of Data Science, Fudan University*

**Zengfeng Huang**[*]                                         *huangzf@fudan.edu.cn*
*School of Data Science, Fudan University*
*Shanghai Innovation Institution*

**Reviewed on OpenReview:** *https://openreview.net/forum?id=SqpgDUdRE9*

## Abstract

Recent research reveals that GNN-based graph representation learning may inadvertently introduce various structural biases. In this work, we discover a phenomenon of structural bias in graph representation learning called class bias amplification, which refers to the exacerbation of performance bias between different classes by GNN encoder. We conduct an in-depth theoretical study of this phenomenon from a novel spectral perspective. Our analysis suggests that structural disparities between nodes in different classes result in varying local convergence speeds for node embeddings. This phenomenon leads to bias amplification in the classification results of downstream tasks. Based on the theoretical insights, we propose random graph coarsening, which is proved to be effective in dealing with the above issue. Finally, we propose an unsupervised graph contrastive learning model called Random Graph Coarsening Contrastive Learning (RGCCL), which utilizes random coarsening as data augmentation and mitigates class bias amplification by contrasting the coarsened graph with the original graph. Extensive experiments on various datasets demonstrate the advantage of our method when dealing with class bias amplification.

## 1 Introduction

Graph representation learning (GRL) aims to generate embedding vectors capturing both the structure and feature information. Graph neural networks (GNNs) are the primary encoder architecture for GRL (Bojchevski & Günnemann, 2018; Zhu et al., 2020; 2021; Zhang et al., 2021; Zheng et al., 2022; Yang et al., 2026), which are often trained with unsupervised graph contrastive objectives. Such methods are called graph contrastive learning (GCL) and exhibit outstanding performance in various downstream tasks. Compared to traditional unsupervised GRL methods(Perozzi et al., 2014; Grover & Leskovec, 2016), the distinctiveness of GCL lies in the GNN encoder's use of message passing. Due to the encoder's use of message passing, the final embeddings are likely to inherit the structural bias in the graph, which may cause undesirable performance

---

[*]Corresponding authors.

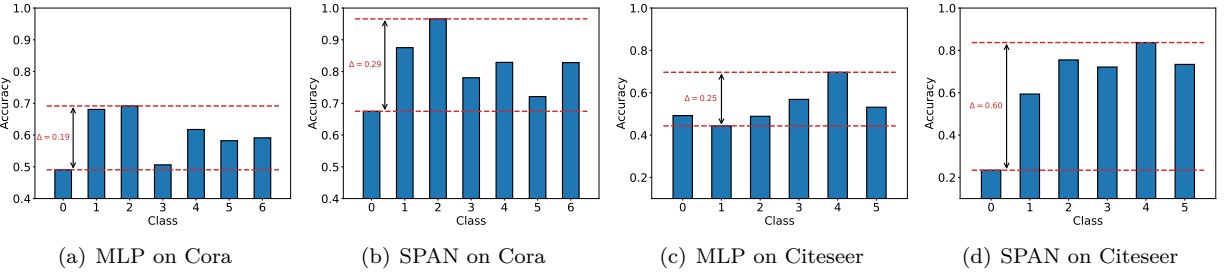

Figure 1: The classification performance of MLP and SPAN in different classes on Cora and Citeseer, where each class has 20 labeled nodes in the training set. $\Delta$ represents the maximum performance difference observed between classes.

unfairness in downstream tasks. This phenomenon is demonstrated in Figure 1, where we compare the node classification performance of MLP (utilizing feature information only) and the state-of-the-art GCL model SPAN (Lin et al., 2023). Although SPAN has a much better overall accuracy than MLP, SPAN exhibits greater performance differences between different classes of nodes. In other words, GNN-based GRL exacerbates the performance bias between different classes.

We refer to the phenomenon exhibited in Figure 1 as *class bias amplification*. This exacerbated bias arises from local structural disparities between nodes in different classes and is unrelated to other information. Graph structural bias problems have been studied in previous works (Tang et al., 2020; Kang et al., 2022; Liu et al., 2023b; Wang et al., 2022). However, they focused on the structure of individual nodes such as degrees and the distance to class boundaries. Class bias amplification studied in this work should also be distinguished from the class imbalance problem (Song et al., 2022). Although both consider the collective bias of class, works on class-imbalanced classification focus on (semi-)supervised learning and aim to reduce prediction bias caused by imbalanced label distributions. However, class bias amplification occurs in GNN-based GRL and is present even when the label distribution is balanced.

This paper investigates the following two questions: *1. Why class bias amplification exists in existing GNN-based GRL methods? 2. Can we design new unsupervised GRL models to alleviate this issue?* To answer the first question, we analyze the structural bias problem from a spectral perspective, which provides a theoretical explanation on the causes of class bias amplification in existing GNN-based GRL models. There have been numerous works conducting spectral analysis of GNNs, e.g., (Kipf & Welling, 2017; Wu et al., 2019; Oono & Suzuki, 2020; Rong et al., 2020). However, existing analyses mainly focus on the global behaviour: They try to characterize the distribution of node representations using the spectrum of the message passing operator. We point out that this is not suitable when structural bias exists across different regions of the graph. If the number of layers in a GNN is not too large as prevalent in real applications, and we define a community as nodes that belong to the same class, then the embedding distributions of different communities are better characterized by their local spectrum. In particular, if the structures of two communities differ a lot, then the second largest eigenvalues of their normalized adjacency matrices can be quite different, which leads to very different convergence speeds to the stationary subspace. As a result, the embedding distributions of the two communities exhibit different levels of concentration. We then show that such an embedding concentration discrepancy can cause unfairness in downstream tasks through a natural statistical model.

Based on the theoretical analysis, we then focus on how to alleviate the class bias amplification. We propose a simple data augmentation technique, namely random graph coarsening, and provide theoretical justifications on the effectiveness. We finally propose an unsupervised graph contrastive learning model, called Random Graph Coarsening Contrastive Learning (RGCCL), which utilizes random graph coarsening as data augmentation and uses a contrastive loss that compares the coarsened graph with the original graph. Empirical results on real datasets show our model effectively reduce performance disparities between different classes and also achieves better overall accuracy than baselines, confirming our theoretical analyses.

Our contributions are summarized as follows:

1. We uncover the class bias amplification in the graph and analyze the causes of this problem from a spectral perspective. We show that local structural bias leads to embedding concentration discrepancy, which is harmful on downstream tasks in terms of fairness.

2. We show that an appropriately designed random graph coarsening algorithm can be used as an effective data augmentation tool for alleviating the issue of embedding density imbalance.

3. Based on our theoretical analysis, we propose a novel GCL model RGCCL. Our model mitigates class bias amplification by comparing the coarsened graph with the original graph.

4. We compare RGCCL with other graph representation learning models in various datasets. Empirical experiments and quantitative analysis demonstrate the advantage of RGCCL, which confirms the effectiveness of using random coarsening to mitigate class bias amplification.

## 2 Preliminaries

**Notation.** Consider an undirected graph $G = (V, E, X)$, where $V$ represents the vertex set, $E$ denotes the edge set, and $X \in \mathbb{R}^{n \times \mathcal{D}}$ is the feature matrix. Let $n = |V|$ and $m = |E|$ represent the number of vertices and edges, respectively. We use $A \in \{0, 1\}^{n \times n}$ to denote the adjacency matrix of $G$ and $\{v_i, v_j\}$ to denote the undirected edge between node $v_i$ and node $v_j$. The degree of node $v_i$ denoted as $d_i$ is the number of edges incident on $v_i$. The degree matrix $D$ is a diagonal matrix and its $i$-th diagonal entry is $d_i$.

**Graph neural network.** In each layer of a GNN, the representation of a node is computed by recursively aggregating and transforming representation vectors of its neighboring nodes from the last layer. One special case is the Graph Convolutional Network (GCN) (Kipf & Welling, 2017). The layer-wise propagation rule of GCN is:

$$H^{(l+1)} = \sigma \left( \widetilde{D}^{-\frac{1}{2}} \widetilde{A} \widetilde{D}^{-\frac{1}{2}} H^{(l)} W^{(l)} \right), \tag{1}$$

where $\widetilde{A} = A + I$, $\widetilde{D} = D + I$ and $W^{(l)}$ is a learnable parameter matrix. GCNs consist of multiple convolution layers of the above form, with each layer followed by an activation $\sigma$ such as Relu.

**Graph coarsening**. The coarse graph is a smaller graph $G' = (A', X')$. $G'$ is obtained from the original graph by computing a partition $P$ of $V$. The partition can be represented by a binary matrix $P \in \{0, 1\}^{n \times n'}$, with $P_{ij} = 1$ if and only if vertex $i$ belongs to cluster $j$. We define $S$ as the set of super-node and for each super-node $i \in S$, $S_i$ as the set of nodes that make up the super-node $i$. $I_u$ is the index of which supernode node $u$ belongs to.

**Convergence of Graph Neural Networks** There has been lots of work investigating the asymptotic behavior of GNNs as the number of layers $L$ goes to infinity, e.g., (Oono & Suzuki, 2020; Rong et al., 2020). The general conclusion is that as $L \to \infty$, the representations of all nodes converge to a 1-dimensional subspace, assuming the graph $G$ is connected. The convergence speed is determined by the second largest eigenvalue of the message passing operator $\hat{A}$. Following the notation from Oono & Suzuki (2020), we denote the maximum singular value of $W^{(l)}$ by $\omega_l$ and set $\omega := \max_{l \in [L]} \omega_l$ and assume that $W^{(l)}$ of all layers are initialized so that $\omega \leq 1$. Given a subspace $\mathcal{M}$, we use $d_{\mathcal{M}} := \inf_{Y \in \mathcal{M}} ||X - Y||_F$ to measure the closeness between $X$ and $\mathcal{M}$, where $|| \cdot ||_F$ denote the Frobenius norm. Oono & Suzuki (2020) shows that if $G$ is connected, there is a 1-d subspace $\mathcal{M}$ such that for all $l$

$$d_{\mathcal{M}}(H^{(l+1)}) \leq \omega \lambda d_{\mathcal{M}}(H^{(l)}), \tag{2}$$

which means the embeddings of all nodes collapse to $\mathcal{M}$ exponentially fast.

## 3 Exploring Class Bias Amplification

In this section, we analyze the reasons for class bias amplification. We initially explore how local structural biases result in discrepancies in embedding concentration. Then, we demonstrate that such discrepancies can cause unfairness in classification tasks.

### 3.1 Illustration of Embedding Concentration Discrepancy

In real applications, the number of layers $L$ is typically small. In these cases, the asymptotic results from Eq.(2) do not provide accurate predictions of the model behavior. In particular, they ignore structural differences between different regions of the graph. Here, we illustrate the problem through a simple example. We consider the following example. There are two classes of nodes in the graph, which form the communities $C_1$ and $C_2$, with all nodes within each community belonging to the same class. $C_1$ is more densely connected than $C_2$, and there are only loose connections between them (see the Figure 2). Then, the expression of the symmetric normalized adjacency matrix $\hat{A}$ can be succinctly represented using a block matrix as follows: $\begin{bmatrix} \hat{A}_1 & \hat{B}_1 \\ \hat{B}_2 & \hat{A}_2 \end{bmatrix}$, where the matrix is partitioned according to $C_1$ and $C_2$. By the assumption, $||\hat{B}_1||_F$ and $||\hat{B}_2||_F$ are close to 0, and since $C_1$ has a better connectivity than $C_2$, the second largest eigenvalue of $\hat{A}_1$, denoted by $\lambda(\hat{A}_1)$ is smaller than $\lambda(\hat{A}_2)$ (Chung & Graham, 1997). According to (2), the representations of all nodes converges to $\mathcal{M}$ with speed exponential in $\lambda(\hat{A})$. A more detailed analysis presented below shows that, the two communities exhibit different convergence speed in the first few layers, due to different local connectivities.

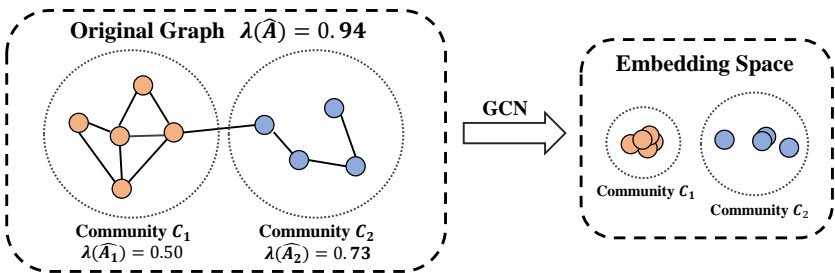

Figure 2: A simple example of embedding density imbalance.

Following the message passing mechanism in Eq.(1), the representations for community $C_1$ and $C_2$ can be defined as $H_1^{(l+1)} = \sigma(\hat{A}_1 H_1^{(l)} W^{(l)} + \hat{B}_1 H_2^{(l)} W^{(l)})$ and $H_2^{(l+1)} = \sigma(\hat{A}_2 H_2^{(l)} W^{(l)} + \hat{B}_2 H_1^{(l)} W^{(l)})$, respectively. Since $||\hat{B}_1||_F$ and $||\hat{B}_2||_F$ are very close to 0, the impact of $H_2^{(l)}$ on $H_1^{(l+1)}$ and $H_1^{(l)}$ on $H_2^{(l+1)}$ is very small. Based on the result of Oono & Suzuki (2020), we have:

$$\begin{cases} d_{\mathcal{M}}(H_1^{(l+1)}) \approx d_{\mathcal{M}}(\sigma(\hat{A}_1 H_1^{(l)} W^{(l)})) \leq \omega\lambda(\hat{A}_1)d_{\mathcal{M}}(H_1^{(l)}), \\ d_{\mathcal{M}}(H_2^{(l+1)}) \approx d_{\mathcal{M}}(\sigma(\hat{A}_2 H_2^{(l)} W^{(l)})) \leq \omega\lambda(\hat{A}_2)d_{\mathcal{M}}(H_2^{(l)}), \end{cases} \tag{3}$$

When the number of iterations $L$ is relatively small as in many real applications, the effects of $\hat{B}_1$ and $\hat{B}_2$ can be ignored, and the embedding distributions of different regions $d_{\mathcal{M}}(H_1^{(L)})$ and $d_{\mathcal{M}}(H_2^{(L)})$ are much better characterized by local connectivity (3). Therefore, the embedding density of each community is primarily determined by nodes that constitute the community.

If the structure of communities $C_1$ and $C_2$ differs dramatically, for example, $\lambda(\hat{A}_1) \ll \lambda(\hat{A}_2)$, then after $L$ layers of message passing, we have $d_{\mathcal{M}}(H_1^{(L)}) \ll d_{\mathcal{M}}(H_2^{(L)})$. This means the embeddings of nodes in $C_1$ will be much more concentrated than those in $C_2$. In other words, the disparity in local convergence speeds leads to an imbalance in local embedding densities. Figure 2 provides an example of this phenomenon. We sample feature vectors for nodes in $C_1$ and $C_2$ from normal distributions, $\mathcal{N}(\begin{bmatrix} -1 \\ -1 \end{bmatrix}\begin{bmatrix} 1 & 0 \\ 0 & 1 \end{bmatrix})$ and $\mathcal{N}(\begin{bmatrix} 1 \\ 1 \end{bmatrix}, \begin{bmatrix} 1 & 0 \\ 0 & 1 \end{bmatrix})$, respectively. We then apply a 2-layer graph convolutional network, and the resulting embeddings of the nodes are visualized. Even though the node features for $C_1$ and $C_2$ are sampled with the same variance, the variance of their embeddings differs due to the convergence bias resulting from the distinct structures.

To further illustrate the differences in the variances of their embeddings, we provide a more quantitative analysis based on the contextual stochastic block model (CSBM) (Deshpande et al., 2018). CSBM is a widely

used statistical model for analyzing expressive power of GNNs (Baranwal et al., 2021; Wu et al., 2022). We consider a two-block CSBM denoted as $\mathcal{G}(n, p_1, p_2, q, \mu_1, \mu_2, \sigma^2)$. Here, $A \in \mathbb{R}^{n \times n}$ represents the adjacency matrix of the graph, and $X \in \mathbb{R}^{n \times d}$ represents the feature matrix. In this model, for any two nodes in the graph, the intra-class probability is denoted as $p_i$ $(i = 1, 2)$, and the inter-class probability is denoted as $q$. Additionally, each node's initial feature is independently sampled from a Gaussian distribution $\mathcal{N}(\mu_i, \sigma^2)$.

Our objective is to estimate the variance of node embeddings within each class. Formally, we aim to compute:

$$\mathbb{E}\left[\left\|D^{-1}AX - \mathbb{E}(D^{-1}AX)\right\|_F^2\right]. \tag{4}$$

**Assumption 3.1** (Structural Information). *$p_1, p_2, q = \Omega(\frac{\log n}{n})$ and $p_1 > p_2 > q$.*

**Lemma 3.1.** *Assume that $d > \frac{C}{\epsilon^2}\log n$, we have $(1 - \epsilon)\|A - \mathbb{E}A\|_F^2 \leq \|AX - \mathbb{E}AX\|_F^2 \leq (1 + \epsilon)\|A - \mathbb{E}A\|_F^2$ with probability at least $1 - 2\exp(-c\epsilon^2 d)$.*

*Proof.* Let $X \in \mathbb{R}^{n \times d}$ be the projection matrix which maps each vector $A_i \in \mathbb{R}^n$ to a $d$-dimensional vector $A_i X \in \mathbb{R}^d$. Then according to the Johnson-Lindenstrauss lemma (Johnson & Lindenstrauss, 1984), given some tolerance $\epsilon$, it holds with probability at least $1 - 2\exp(-c\epsilon^2 d)$. □

**Lemma 3.2.** *For a given Erdős-Rényi (ER) graph $\mathcal{G}(n, p)$, there exists a constant $C$ such that $\|A - \mathbb{E}A\| \lesssim C\sqrt{np}$ with probability at least $1 - n^{-r}$ for any $r > 0$.*

*Proof.* By using corollary 3.12 from (Bandeira & van Handel, 2016), we obtain sharper bounds. □

**Lemma 3.3** (Sharp concentration, Lemma 3.2 and Theorem 3 from (Wu et al., 2022)). *There exists a constant $C$ such that for sufficiently large $n$, with probability at least $1 - O(n^{-r})$,*

$$\left\|D^{-1}A - \mathbb{E}(D^{-1}A)\right\|_F \lesssim \frac{C}{\sqrt{np}}. \tag{5}$$

**Theorem 3.4.** *Given an CSBM $\mathcal{G}(n, p_1, p_2, q, \mu_1, \mu_2, \sigma^2)$, it holds that for the variance of class I with intra-class probability $p_1$ is smaller than the the variance of class II with intra-class probability $p_2$.*

*Proof.* According to Lemma 3.1, to measure the variance of node embedding, we just need to consider structure influence $\left\|D^{-1}A - \mathbb{E}(D^{-1}A)\right\|_F^2$. By expressing $A = \begin{bmatrix} A_1 & B \\ B & A_2 \end{bmatrix}$, this allows us to focus on ER graph $\mathcal{G}(\frac{n}{2}, p_i)$ for each class separately. Denote $R_1 = D^{-1}A_1 - \mathbb{E}(D^{-1}A_1)$ and $R_2 = D^{-1}A_2 - \mathbb{E}(D^{-1}A_2)$. We have

$$\|R_1\|_F^2 \leq (1 + \epsilon_1)\frac{C}{np_1} \leq (1 - \epsilon_2)\frac{C}{np_2} \leq \|R_2\|_F^2 \tag{6}$$

for appropriate $\epsilon_1, \epsilon_2$. It follows that variance of node embedding decreases more rapidly for the denser class. Furthermore, the convergence speed is inversely proportional to the intra-probability $p_i$. □

## 3.2 From Embedding Concentration Discrepancy to Class Bias Amplification

We have discussed how different local convergence speeds lead to varying degrees of dispersion in local embedding distributions, and next we provide a theoretical justification on why this can lead to class bias amplification in downstream tasks.

To illustrate the issue, we consider a binary classification problem. Following the setting in Section 3.1, the graph contains two classes of nodes, which form the distinct communities $C_1$ and $C_2$. We assume that the node embeddings from each community follow a Gaussian distribution. We consider the optimal Bayes classifier for the above model, which is known to be the *quadratic discriminant* rule.

**Definition 1** (Quadratic Discriminant Analysis). *For a binary classification problem, where class 1 has mean $\mu_1$ and variance $\Sigma_1$ and class 2 has mean $\mu_2$ and variance $\Sigma_2$. Each sample is drawn from either one with equal probability. Given a new sample $x$, the QDA rule is*

$$\arg\max_{c=1,2} -\frac{1}{2}\log\det\Sigma_c - \frac{1}{2}(x - \mu_c)^T\Sigma_c^{-1}(x - \mu_c). \tag{7}$$

To simplify the discussion, we focus on the one-dimensional case, while the results for higher dimensions are similar. More specifically, each sample $x$ is drawn from $\mathcal{N}(\mu_1, \sigma_1^2)$ or $\mathcal{N}(\mu_2, \sigma_2^2)$ with equal probability, and let $y$ be the label of $x$. Assume $\sigma_1 \neq \sigma_2$, meaning the data distribution of one class is more concentrated than the other, which models the embedding density imbalance. For this simple case, the optimal classification error can be represented as a closed form.

**Proposition 1.** *For the above QDA classifier, the error probability of samples from class 1 is:*

$$p_1 = \mathbb{P}(Y^2 > \frac{(\sigma_1^2 + \sigma_2^2)^2}{\sigma_2^2 - \sigma_1^2} - (\sigma_2^2 - \sigma_1^2) + 2\sigma_1^2\sigma_2^2\log(\frac{\sigma_2}{\sigma_1})), \tag{8}$$

*where $Y \sim \mathcal{N}(\sqrt{|\sigma_2^2 - \sigma_1^2|} + (2I(\sigma_1 > \sigma_2) - 1)\frac{\sigma_1^2 + \sigma_2^2}{\sqrt{|\sigma_2^2 - \sigma_1^2|}}, |\sigma_1^2 - \sigma_2^2|\sigma_1^2).$*

The classification error probability of class 2 is symmetric and thus omitted. The above expression is quite complicated. To demonstrate the class bias amplification issue, we define the degree of class bias as $\kappa = \frac{\max\{p_1, p_2\}}{\min\{p_1, p_2\}}$ (larger $\kappa$ means more severe class bias issue), and investigate how the imbalance in data distribution affects $\kappa$. To this end, we fix the value of $\sigma_1^2 + \sigma_2^2$, and vary the ratio $\sigma_1/\sigma_2$.

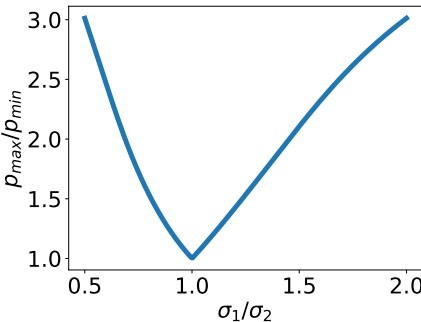

Figure 3: Fairness $\kappa$ vs. ratio between variances.

The value of $\kappa$ corresponding to different ratios is plotted in Figure 3. It is evident that as the degree of imbalance in $\sigma_1/\sigma_2$ increases, $\kappa$ also increases. That is to say, as the degree of imbalance in embedding density grows, the class bias gradually increases as well. Overall, local structural differences exacerbate the embedding concentration discrepancy, which further leads to the amplification of class bias in the classifier.

## 4 Random Graph Coarsening Contrastive Learning

In this section, we first present the main idea and theoretical justifications of random coarsening to alleviate the class bias amplification. Additionally, we propose the Random Graph Coarsening Contrastive Learning (RGCCL) model, which utilizes random graph coarsening as data augmentation.

### 4.1 Mitigating Class Bias Amplification via Random Coarsening

Our goal is to make the embedding distribution of sparse classes more concentrated. For this to happen, we first randomly partition the graph into clusters $S = \{S_1, \cdots, S_t\}$ according to some random process. Let $f$ be a GNN encoder, and $f(u)$ be the embedding of node $u$. We define the embedding for a cluster $S_i$ as $f(S_i) = \frac{1}{|S_i|} \sum_{v \in S_i} f(v)$. We use the loss

$$\sum_{S_i \in S} \sum_{u \in S_i} \|f(u) - f(S_i)\|^2 \tag{9}$$

to regularize the GNN encoder, which encourages nodes in each cluster in the random partition to be more concentrated. In the following, we first show that if the distribution of the random partition, denoted

by $\mathcal{P}$, satisfies certain requirements, the above loss has the implicit effect of pushing the embeddings of sparse classes more heavily. Then, we provide a specific random partition algorithm, namely random graph coarsening, which meets the requirements.

Following the setting in Section 3.2, we consider a binary classification problem. Assume that for a random partition drawn from $\mathcal{P}$, the probability that two nodes from class 1 (class 2) lie in the same cluster is $q_1$ ($q_2$), and the probability that two nodes from different classes are clustered together is $q_{12}$. We have the following lemma, the proof of which is provided in the appendix A.1.

**Lemma 4.1.** *Let $C_1$ and $C_2$ be the two classes of nodes. Suppose each cluster has the same size $s$, then*

$$
\mathbb{E}_{P\sim\mathcal{P}} \sum_{S_i \in S} \sum_{u \in S_i} s\|f(u) - f(S_i)\|^2
$$
$$
=q_1 \sum_{u,v \in C_1, u \neq v} \|f(u) - f(v)\|^2 + q_2 \sum_{u,v \in C_2, u \neq v} \|f(u) - f(v)\|^2
$$
$$
+ q_{12} \sum_{u \in C_1, v \in C_2} \|f(u) - f(v)\|^2.
$$

Now suppose $C_1$ is denser than $C_2$, and by the analysis in Section 3.1, the embeddings in $C_1$ are likely to be more concentrated. To make the embedding densities of $C_1$ and $C_2$ more balanced, from Lemma 4.1, a preferred random partitioning algorithm should satisfy

$$
q_2 > q_1 > q_{12}. \tag{10}
$$

Here, we design a random graph coarsening strategy to obtain such a reasonable partition, which can satisfy (10). Due to the homophily principle that similar nodes may be more likely to attach to each other than dissimilar ones, if we always merge nodes that are connected by an edge, $q_{12}$ should be less than $q_1$ and $q_2$. On average, the degrees of nodes in $C_2$ are lower than $C_1$. To realize $q_2 > q_1$, we adopted a simple probability function $\omega(u, v) = \frac{1}{d_u + d_v}$ for random edge selection during coarsening, ensuring a higher probability for low-degree nodes to participate in the random coarsening. Specifically, we first compute $\omega(u, v)$ to construct the weight set $\mathcal{I}$ of all edges, and then determine the edges to be coarsened by randomly selecting edges according to the weight set $\mathcal{I}$.

Therefore, our random graph coarsening strategy iteratively merging two (super) nodes through edge contraction to form more super nodes. Eventually, we can obtain a coarsened graph, where each supernode represents a cluster, and the nodes that constitute this supernode belong to the same partition. By contrasting the nodes in the original graph with their corresponding supernodes (clusters) in the coarsened graph, we can optimize loss (9) to mitigate class bias amplification. To prevent the formation of large supernodes, we use a threshold limiting the size of supernodes during the coarsening process. A detailed description of our random coarsening algorithm is provided in the appendix A.4.

## 4.2 Framework of RGCCL

Based on the theoretical insights from Section 4.1, we present a novel multi-view graph contrastive learning method RGCCL, which can effectively alleviate the issue of class bias amplification. Compared to existing GCL models, the key difference in our approach is to use random graph coarsening as the graph augmentation method and a specially designed loss function for this framework.

The GCL method generates different views through graph augmentation, and then trains the model parameters by comparing the node embeddings from different views. The architecture of RGCCL is presented in Figure 4. In our RGCCL, we regard the original graph as one view and the coarsened graph as another view, and then compare the corresponding nodes in these two views. In order to alleviate class bias amplification, we need to push the node embedding $f(u)$ towards its corresponding cluster center embedding $f(S_u)$ according to the analysis in Section 4.1. Specifically, the super-node embedding computed in the coarsened view is used as the cluster center, and then each node embedding in the original graph and its corresponding

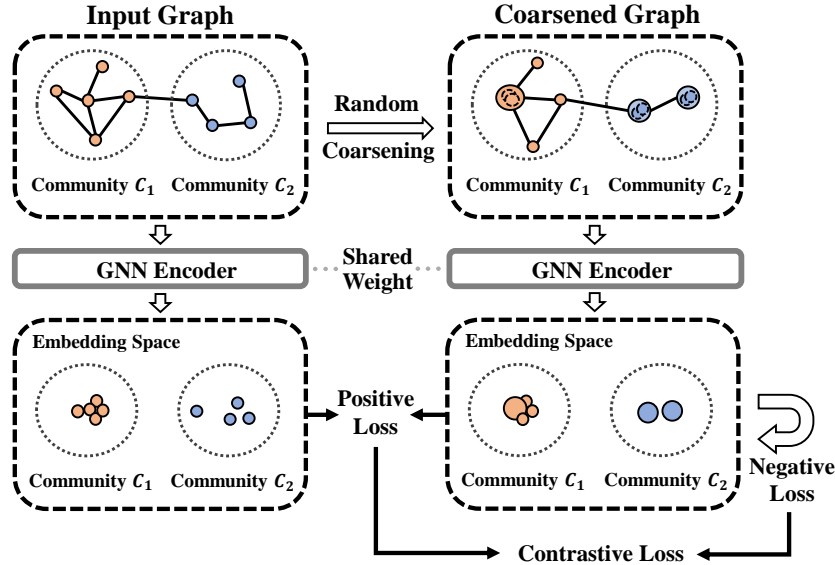

Figure 4: The architecture of the proposed RGCCL.

super-node embedding in the coarsened graph is defined as a positive pair. By conducting contrastive learning on such positive pairs, we make the embedding distribution of sparse classes more concentrated, thereby reducing class bias amplification.

In this work, we use a method different from the previous graph coarsening algorithms (Huang et al., 2021) to construct the coarsened feature matrix. A difference is that they construct the new feature matrix simply by summing i.e., given a partition matrix $P$, $X' = P^T X$. Here, a normalization based on degrees is applied: $X' = \widetilde{D}'^{-1} P^T \widetilde{D} X$, where $\widetilde{D}'$ and $\widetilde{D}$ are the degree matrices of the two views. The purpose is to ensure, as the number of graph propagations goes to infinity, the embedding of a node and its corresponding supernode in the coarsened view converge to the same point. This computational method is supported by the following upper bound

$$\|Z_u - Z'_v\| \le \kappa \sqrt{\frac{d_{max}}{d_{min}}} (n\lambda_2^k + n'\lambda_2'^k). \tag{11}$$

Here, $u$ represents a node in the original graph, while $v$ is its corresponding node in the coarsened graph. We denote the node embeddings learned from the original graph $G$ and the coarsened graph $G'$ as $Z_u$ and $Z'_v$, respectively. $\lambda_2$ and $\lambda'_2$ are the second largest eigenvalues in $G$ and $G'$ respectively. $d_{max}$ and $d_{min}$ are the maximum degree and minimum degree in $G$ and $G'$, respectively. A detailed proof can be found in Appendix A.2.

On the other hand, graph coarsening is a method of data reduction, so using random graph coarsening as data augmentation can also reduce the resource consumption of GNN training, especially when computing positive and negative sample pairs.

Specifically, we apply the random graph coarsening algorithm to generate one graph augmentation $G' = (A', X')$ in each epoch during training. Then, we compute the coarsened graph and original graph embeddings by a GNN encoder with shared parameters: $H = \text{GNN}(A', X', \theta)$ and $Z = \text{GNN}(A, X, \theta)$.

Recall that positive pairs are of the form $(u, S_u)$, where $S_u$ denotes the corresponding supernode of $u$. So we penalize $\|Z_u - H_{S_u}\|_F^2$. The loss function of positive pairs can be described more concisely in the matrix form. Let $Z' = PH$, then $Z'_u = H_{S_u}$. Therefore, the positive pair loss function is

$$\mathcal{L}_{pos} = \|Z - Z'\|_F^2 = \|Z\|_F^2 + \|Z'\|_F^2 - 2\text{Tr}(Z^T Z'). \tag{12}$$

If $Z$ and $Z'$ are normalized appropriately, we only need to minimize $-2\text{Tr}(Z^T Z')$.

Considering that the node embeddings $Z$ derived from the original graph $G$ may encounter imbalance issues, whereas the coarsened graph $G'$ typically exhibits a more balanced embedding distribution. Therefore, we do not pick negative pairs from the original graph and only compute a negative pair loss with the coarsened graph. There are various methods for selecting negative pairs and computing the loss; here we use the loss from Zhang et al. (2020), which is derived from the graph partition problem. More specifically, we randomly sample a small set of supernode pairs $\mathcal{N}' \subset V' \times V'$, and the negative pair loss function is:

$$\mathcal{L}_{neg} = \frac{\alpha}{\sum_{(i,j)\in\mathcal{N}'} n_i n_j \|h_i - h_j\|^2}. \tag{13}$$

where $h_i$ and $h_j$ are the embeddings of supernodes $i$ and $j$, and $n_i$ and $n_j$ are the number of original nodes contained in supernodes $i$ and $j$.

Optimizing $\mathcal{L}_{neg} + \mathcal{L}_{pos}$ will be difficult due to the huge difference of the scale of $\mathcal{L}_{pos}$ and $\mathcal{L}_{neg}$. Therefore, we transform $\mathcal{L}_{pos}$ into the following form

$$\mathcal{L}_{pos} = \frac{\beta}{\mathsf{Tr}(Z^T Z')}. \tag{14}$$

Finally, the loss function of our model is

$$\mathcal{L} = \frac{\alpha}{\sum_{(i,j)\in\mathcal{N}'} n_i n_j \|h_i - h_j\|^2} + \frac{\beta}{\mathsf{Tr}(Z^T Z')}. \tag{15}$$

### 4.3 Generalizability of RGCCL

The generalizability of self-supervised learning methods has recently been theoretically analyzed in (Huang et al., 2023; Wang et al., 2022). They characterize it with three properties, namely the concentration of augmented data, the alignment of positive samples and the divergence of class centers. Huang et al. (2023) also shows that the divergence of class centers is controlled by classic contrastive losses such as InfoNCE and the cross-correlation loss.

Following the proof of Huang et al. (2023), we show that our self-supervised loss $\mathcal{L}_{neg}$ can also upper bounds the divergence of class centers, thus classes will be more separable if our objective is optimized. The theorem is stated as follows, with further details provided in Appendix A.3.

**Theorem 4.2.** *Assume that encoder $f$ with norm $1$ is $M$-Lipschitz continuous. For a given $(\alpha, \gamma, \hat{d})$ augmentation set $A$, any $\epsilon > 0$ and $k \neq l$,*

$$\mu_k^T \mu_l \leq \frac{1}{p_k p_l}\left(-\frac{1}{2n^2 \mathcal{L}_{neg}} + \tau(\epsilon, \alpha, \gamma, \hat{d})\right), \tag{16}$$

*where* $\tau(\epsilon, \alpha, \gamma, \hat{d}) = 2R_\epsilon + 16(1-\alpha(1-\frac{1}{2}\epsilon-\frac{1}{4}M\max_k(\gamma\sqrt{\frac{\mathcal{D}}{\hat{d}_{\min}^k}}))+KR_\epsilon)^2 + 8(1-\alpha(1-\frac{1}{2}\epsilon-\frac{1}{4}M\max_k(\gamma\sqrt{\frac{\mathcal{D}}{\hat{d}_{\min}^k}}))+KR_\epsilon + \frac{K-1}{K})$.

Next we discuss the concentration property of the random coarsening augmentation. First notice that for any two nodes $u$ and $v$, the probability that they are coarsened together is equal to the probability that they are connected by randomly selected edges in the algorithm. Suppose the edges are selected independent and identically with probability $p$, then the connection probability is lower bounded by $p^{\mathrm{dia}(G)}$, where $\mathrm{dia}(G)$ is the diameter of $G$. Once $u$ and $v$ are coarsened together in at least one coarsened graph, $d(u,v) = 0$, which means our random coarsening augmentation can be very well-concentrated.

## 5 Related Work

**Contrastive learning on graphs.** Contrastive learning is a type of unsupervised learning technique that learns a representation of data by differentiating similar and dissimilar samples. It has been used

in a variety of applications within the domain of graph data. DGI (Veličković et al., 2018) and MVGRL (Hassani & Ahmadi, 2020) contrast node embeddings with graph embeddings using a loss function based on mutual information estimation (Belghazi et al., 2018; Hjelm et al., 2019). GRACE (Zhu et al., 2020) and its variants (Zhu et al., 2021; You et al., 2020) aim to maximize the similarity of positive pairs and minimize the similarity of negative pairs in augmented graphs in order to learn node embeddings. To counter the performance degradation induced by false negative pairs, CCA-SSG (Zhang et al., 2021) simplifies the loss function by eliminating negative pairs. In order to reduce the computational complexity of contrastive loss, GGD (Zheng et al., 2022) discriminates between two groups of node samples using binary cross-entropy loss. For a broader understanding, we recommend that readers refer to the latest surveys (Liu et al., 2023a; Xie et al., 2023).

**Structural bias on graphs.** Fair graph mining has attracted much more research attention since recent studies reveal that there are unfairness in a large number of graph mining models. Several notions of fairness have been proposed in recent survey (Dong et al., 2023), and structural bias is mainly manifested as degree bias. Prior studies (Tang et al., 2020; Kang et al., 2022; Dong et al., 2022; Liu et al., 2023b) have primarily concentrated on degree bias in supervised graph learning. GRADE (Wang et al., 2022) first focuses on structural bias in unsupervised graph representation learning, mitigating the degree bias issue through a degree-based graph data augmentation method. In view of the wide application of GNN-based graph representation learning, the structural bias problem associated with it should receive more attention from researchers. Existing studies on bias amplification, such as those in computer vision and NLP (Zhao et al., 2017; Wang & Russakovsky, 2021; Hall et al., 2022), have not considered graph structural bias, and their definitions of bias amplification are primarily based on output distribution shifts rather than structural propagation.Different from them, we are the first to study the phenomenon of class bias amplification on graphs, which specifically captures how initial class imbalances are progressively exacerbated by GNN structures, rather than demographic or attribute biases commonly discussed in prior works.

**Graph coarsening.** Recently, graph coarsening techniques have been used to address issues in graph neural networks (Fahrbach et al., 2020; Deng et al., 2019; Huang et al., 2021; Jin et al., 2022; Yang et al., 2025). Graph coarsening with spectral approximation guarantees are studied in (Li & Schild, 2018; Loukas, 2019; Jin et al., 2020). Graph coarsening can reduce the size of graph by combining the similar nodes, then the coarsened graph can be used for downstream tasks related to the graph. Existing graph coarsening techniques primarily strive to maintain the overall graph structure, resulting in a static and downsized coarsened graph. These methods overlooks the local structural bias, and typically involves massive computational costs.

# 6 Experiments

## 6.1 Experimental Setup

**Datasets.** The results are evaluated on six real-world datasets (Kipf & Welling, 2017; Veličković et al., 2018; Zhu et al., 2021; Hu et al., 2020), Cora, Citeseer, Pubmed, Amazon Computer, Amazon Photo, and Ogbn-Arixv. Graph representation learning shows different degrees of class bias in these datasets. More detailed statistics of the six datasets are summarized in Table 7 . On small-scale datasets Cora, Citeseer, Pubmed, Photo and Computers, performance is evaluated on random splits. We select 20 labeled nodes per class for training, while the remaining nodes are used for testing. For Ogbn-Arixv, we use fixed data splits as in previous studies Hu et al. (2020).

**Baselines.** We compare our approach against ten representative graph embedding models: Deepwalk (Perozzi et al., 2014), DGI (Veličković et al., 2018), GraphCL (You et al., 2020), GRACE (Zhu et al., 2020), GCA (Zhu et al., 2021), CCA-SSG (Zhang et al., 2021), gCool (Li et al., 2022), GRADE(Wang et al., 2022) , GGD(Zheng et al., 2022), SPAN(Lin et al., 2023) and PolyGCL(Chen et al., 2024). For all the baselines, we use the public code released in their previous papers. All models evaluate the learned representations by training and testing classifiers with the same settings. More experimental details are listed in the appendix A.5.

## 6.2 Results and Analysis

Table 1 reports the node classification performance on small datasets. In these data splits, we employ Acc and Macro-F1 as metrics to assess the overall performance of models. If Macro-F1 significantly drops compared to Acc, it indicates that the model's performance is not balanced across different classes. It is evident that the performance of RGCCL outperforms other GRL models within the given experimental framework. Mostly, RGCCL surpasses the runner-up by an advantage of 1%-2%. Notably, although RGCCL's Acc score on the Computers dataset is slightly lower than that of CCA-SSG, our Macro-F1 score is significantly higher than that of CCA-SSG. This suggests that CCA-SSG has generated significant class bias on Computers. Clearly, our specially designed model can improve the overall classification performance effectively while resolving the issue of class bias amplification.

Table 1: Summary of results in terms of mean node classification accuracy and standard deviation over 50 runs on five datasets.

| Method | Cora | | Citeseer | | Pubmed | | Photo | | Computers | |
|---|---|---|---|---|---|---|---|---|---|---|
| | Acc | Macro-F1 | Acc | Macro-F1 | Acc | Macro-F1 | Acc | Macro-F1 | Acc | Macro-F1 |
| Deepwalk | 67.2±1.7 | 66.5±1.5 | 40.0±2.1 | 38.3±2.0 | 66.9±2.8 | 65.6±2.7 | 85.1±1.2 | 83.9±1.2 | 77.3±1.6 | 77.2±1.5 |
| DGI | 78.5±0.9 | 77.2±0.9 | 70.4±1.0 | 63.6±1.4 | 72.5±3.3 | 72.5±3.3 | 87.9±1.3 | 86.2±1.3 | 79.7±1.6 | 78.4±1.3 |
| GraphCL | 78.3±1.5 | 76.7±1.7 | 70.6±1.2 | 64.1±1.4 | 71.7±3.5 | 71.7±3.6 | 88.3±1.3 | 86.7±1.2 | 79.7±1.4 | 78.5±1.1 |
| GRACE | 74.4±2.0 | 72.5±2.0 | 68.9±1.0 | 61.2±1.1 | 76.1±2.8 | 75.9±2.7 | 85.1±1.6 | 83.5±1.4 | 76.2±1.9 | 75.2±1.5 |
| GCA | 78.6±1.2 | 77.2±1.2 | 68.8±1.5 | 65.3±1.4 | 75.4±3.0 | 75.5±2.9 | 87.8±1.2 | 86.2±1.3 | 79.1±2.4 | 77.9±2.0 |
| CCA-SSG | 79.2±1.4 | 78.0±1.4 | 71.8±1.0 | 66.3±1.1 | 76.0±2.8 | 75.8±2.7 | 88.7±1.1 | 86.9±3.2 | 82.7±1.0 | 76.9±3.7 |
| gCooL | 78.5±1.3 | 77.1±1.1 | 68.6±1.4 | 64.9±1.3 | 75.5±3.0 | 75.3±2.9 | 87.9±1.3 | 85.9±1.4 | 79.8±1.7 | 78.1±1.3 |
| GRADE | 81.5±1.0 | 80.2±1.0 | 67.6±1.5 | 64.2±1.3 | 74.5±2.7 | 74.5±2.6 | 87.1±1.2 | 80.4±3.0 | 75.8±1.2 | 64.7±3.1 |
| GGD | 81.9±0.9 | 80.5±0.8 | 70.1±1.3 | 66.2±1.1 | 74.7±3.2 | 74.4±3.1 | 87.2±1.5 | 85.4±1.4 | 80.4±1.8 | 80.0±1.2 |
| SPAN | 80.7±0.6 | 78.8±0.7 | 69.1±1.2 | 64.3±1.1 | 73.1±2.8 | 72.6±2.8 | 86.3±1.5 | 84.4±1.6 | 76.4±1.7 | 75.8±1.4 |
| PolyGCL | 77.4±3.2 | 75.8±3.1 | 69.9±1.4 | 62.7±1.3 | 75.2±3.4 | 75.5±3.1 | 85.7±1.4 | 84.6±1.3 | 76.7±1.6 | 75.5±1.5 |
| RGCCL | **83.1±0.8** | **82.0±0.8** | **72.4±0.9** | **67.7±0.8** | **77.3±2.9** | **77.1±2.7** | **89.6±1.2** | **88.2±1.2** | 81.2±1.8 | **80.2±1.2** |

**Quantitative analysis.** To further demonstrate that RGCCL effectively alleviates the amplification of class bias, we present statistics of the learned representations and compare them with other popular GRL methods. Firstly, we measure the concentration of the embedding for each class by calculating their mean distances from the centroid (i.e., $V_C = \frac{1}{|C|} \sum_{i \in C} ||z_i - \frac{1}{|C|} \sum_{i \in C} z_i||$ for class $C$). Next, we compute the average and standard deviation of $V_C$ across all classes. A smaller average value indicates that the representations for each class are more concentrated, which in turn makes the classification boundary easier to learn. A smaller standard deviation suggests a more balanced embedding density, which has been shown to be beneficial for classification fairness (as discussed in Section 3.2). In Table 2, we present the results of four baseline methods as well as RGCCL on Cora and Citeseer. RGCCL demonstrates not only more concentrated embeddings for each class but also the most balanced embedding density. These results strongly support our theory. Furthermore, we use the Matthew's coefficient to measure class bias. Table 3 presents the results of Matthew's coefficient for representative GRL models and our RGCCL. The results indicate that the bias in the embeddings learned by RGCCL is significantly less than that of other GRL models.

**Visualization.** To further illustrate that RGCCL effectively mitigates the issue of class bias amplification, we have visualized the performance of each model across different classes. For each class, we compute the

Table 2: The average and standard deviation of class density. A smaller average indicates higher embedding quality, while a smaller standard deviation suggests less class bias.

| Method | Cora | | Citeseer | |
|---|---|---|---|---|
| | Ave | Std | Ave | Std |
| DGI | 0.3782 | 0.0294 | 0.3402 | 0.0255 |
| GRACE | 0.2114 | 0.0252 | 0.1715 | 0.0163 |
| CCA-SSG | 0.2817 | 0.1031 | 0.1672 | 0.0109 |
| GRADE | 0.1983 | 0.0352 | 0.2141 | 0.0243 |
| GGD | 0.3183 | 0.0541 | 0.3297 | 0.0329 |
| SPAN | 0.3260 | 0.0323 | 0.3234 | 0.0227 |
| RGCCL | **0.0942** | **0.0084** | **0.1401** | **0.0097** |

Table 3: Matthew's coefficient for RGCCL and six baselines.

|  | Cora | CiteSeer | PubMed |
|---|---|---|---|
| DGI | 73.0±1.5 | 64.5±1.2 | 57.8±4.3 |
| GRACE | 67.9±1.6 | 60.0±2.0 | 62.3±3.8 |
| CCA-SSG | 74.5±1.6 | 64.6±1.4 | 64.0±4.1 |
| GRADE | 75.7±1.5 | 62.1±1.3 | 58.1±3.4 |
| GGD | 77.0±1.6 | 64.9±1.2 | 63.7±4.1 |
| SPAN | 76.9±0.7 | 62.4±1.4 | 60.3±4.0 |
| RGCCL | **78.9**±0.9 | **66.3**±0.8 | **65.6**±4.2 |

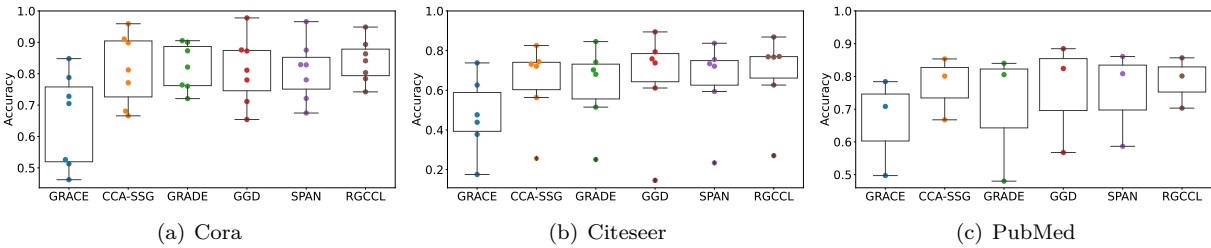

(a) Cora  (b) Citeseer  (c) PubMed

Figure 5: Box plots of the average accuracy w.r.t. class for five baselines and RGCCL on the Cora, Citeseer and Pubmed dataset.

average performance of the model in this class and then draw a box plot based on these accuracies. Figure 5 provides a visualization of the node classification accuracy in different classes on the Cora, Citeseer and PubMed. It is clear that our model has the smallest performance difference across different classes, while also having the best overall performance.

**Scalability.** Another benefit of RGCCL is its small memory usage. This efficiency is primarily because the random graph coarsening is preprocessed on the CPU, resulting in a coarsened graph notably smaller than the original. Experiments were also conducted on the Arxiv dataset, which is a large-scale dataset for most GRL models. Larger size makes sub-sampling necessary for training on some baselines, but our RGCCL can be trained directly on the full graph. As shown in Table 4, the Acc and Macro-F1 of RGCCL both exceed those of other baselines.

Figure 6 shows the memory usage of our model and seven mainstream GCL models on Citeseer and Pubmed. RGCCL exhibits the same level of memory usage as GGD, which is specifically designed to save computation costs; our model benefits from the effective reduction of graph size via random coarsening.

**Effectiveness of different coarsening ratios.** We studied the effect of random coarsening ratio on model performance. The random coarsening ratio refers to the proportion of the number of nodes reduced relative to the total number of nodes. Table 5 shows the results of different coarsening ratios. According to our

Table 4: Summary of results in terms of accuracy on Ogbn-Arxiv.

| Method | Ogbn-Arxiv | |
|---|---|---|
|  | Acc | Macro-F1 |
| DGI | 68.8±0.2 | 46.8±0.4 |
| GRACE | 68.4±0.1 | 46.1±0.3 |
| CCA-SSG | 69.8±0.2 | 46.5±0.6 |
| GRADE | 67.7±0.2 | 45.0±0.4 |
| GGD | 70.7±0.3 | 48.5±0.4 |
| SPAN | 70.1±0.3 | 48.7±0.5 |
| RGCCL | **71.7**±0.1 | **50.6**±0.2 |

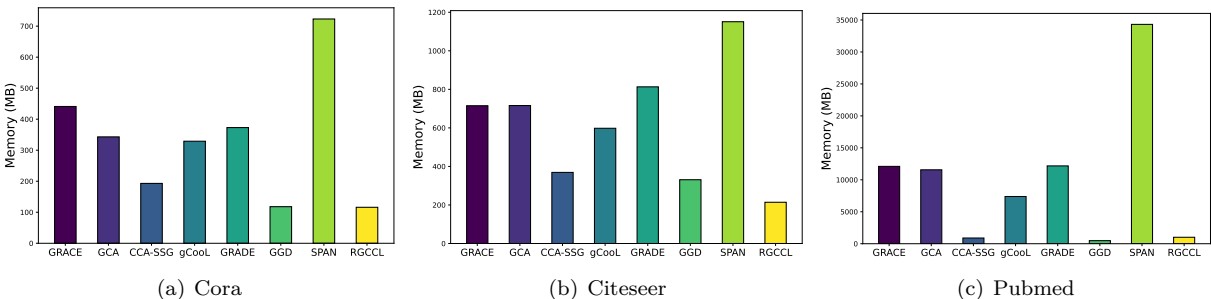

Figure 6: The memory usage of baselines and RGCCL on Cora, Citeseer and Pubmed.

observations, the changes in the coarsening rate have a more significant impact on the Macro-F1. The model performs better when the coarsening rate is between 30% and 50%. If the coarsened graph is too small, it may lead to the loss of information in the augmented graph, which could in turn cause a decline in the performance of the RGCCL.

Table 5: The performance of different coarsening ratio.

| Ratio | Cora | | Citeseer | | Pubmed | |
|---|---|---|---|---|---|---|
| | Acc | Macro-F1 | Acc | Macro-F1 | Acc | Macro-F1 |
| $r = 0.3$ | **83.1**±0.8 | **82.0**±0.8 | **72.4**±0.9 | 67.6±0.8 | 77.1±2.9 | 76.9±2.7 |
| $r = 0.5$ | 82.8±0.8 | 81.8±0.8 | **72.4**±0.9 | **67.7**±0.8 | **77.3**±2.9 | **77.1**±2.7 |
| $r = 0.7$ | 82.5±0.9 | 81.4±0.9 | 72.1±0.7 | 67.0±0.7 | 77.1±2.8 | 76.8±2.6 |
| $r = 0.9$ | 82.4±0.9 | 81.1±1.0 | 72.0±0.7 | 66.9±0.7 | 77.0±2.8 | 76.9±2.6 |

Table 6: The performance of different loss terms ratio.

| Method | Cora | | Citeseer | | Pubemd | |
|---|---|---|---|---|---|---|
| | Acc | Macro-F1 | Acc | Macro-F1 | Acc | Macro-F1 |
| RGCCL | **83.1**±0.8 | **82.0**±0.8 | **72.4**±0.9 | **67.7**±0.8 | **77.3**±2.9 | **77.1**±2.7 |
| RGCCL w/o $\mathcal{L}_{neg}$ | 80.8±0.9 | 79.8±0.8 | 58.0±3.4 | 54.6±2.9 | 67.7±6.0 | 66.9±6.8 |
| RGCCL w/o $\mathcal{L}_{pos}$ | 82.8±0.8 | 81.7±0.7 | 72.0±0.8 | 67.4±0.7 | 76.6±2.8 | 76.4±2.6 |

**Effectiveness of different loss terms.** We conducted an ablation study to evaluate the contribution of different loss terms in our model. Specifically, we examined the impact of removing the positive loss term $\mathcal{L}_{pos}$ and the negative loss term $\mathcal{L}_{neg}$ on node classification performance. As shown in Table 6, the removal of $\mathcal{L}_{neg}$ leads to a significant drop in performance across all three datasets, especially on Citeseer, where the accuracy and Macro-F1 decrease by about 14% and 13%, respectively. This indicates that $\mathcal{L}_{neg}$ plays a crucial role in preserving discriminative information during training. On the other hand, removing $\mathcal{L}_{pos}$ only causes a slight performance degradation, suggesting that while beneficial, its contribution is less critical compared to $\mathcal{L}_{neg}$. Overall, the full model achieves the best results, demonstrating the complementary effects of both loss terms.

# 7 Limitations and Future Work

It is important to emphasize that this study is conducted under the homophilic graph setting, which aligns closely with the mainstream research direction in the field of graph representation learning. This setting ensures that our theoretical framework and methods are well-suited to the widely adopted paradigm—namely, that connected nodes are more likely to share similar attributes. While our method demonstrates excellent performance and robustness under the homophilic assumption, its applicability to heterophilic graphs remains an open question. Heterophilic graphs are characterized by connections between dissimilar nodes, presenting unique challenges that are beyond the scope of this study. In future work, we plan to extend the exploration

of class bias amplification to heterophilic graph settings. This direction will likely require the development of new theoretical tools and the establishment of evaluation metrics tailored to the characteristics of heterophilic graphs.

## 8 Conclusion

In this paper, we study the issue of class bias amplification in GNN-based graph representation learning. We present a novel perspective on this problem through the lens of convergence bias and embedding concentration discrepancy, and a comprehensive theoretical analysis is provided. Based on our theoretical insights, we propose to use random graph coarsening to mitigate this issue, and give theoretical guidance on how to design effective random coarsening algorithms. Finally, a graph contrastive learning model is proposed which utilize random graph coarsening as graph augmentation and a loss function is designed for this new form of graph augmentation. Comprehensive experimental evaluation illustrates the superiority of RGCCL in mitigating class bias amplification.

## Acknowledgments

This work is supported by National Natural Science Foundation of China No. U2241212, No. 62276066, and the Postdoctoral Fellowship Program of CPSF under Grant No. GZC20252742. The computations in this research were performed using the CFFF platform of Fudan University.

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

# A  Appendix

## A.1  The proof of Lemma 4.1

**Lemma A.1.** *Let $C_1$ and $C_2$ be the two classes of nodes. Suppose each cluster has the same size $s$, then*

$$\mathbb{E}_{P \sim \mathcal{P}} \sum_{S_i \in S} \sum_{u \in S_i} s \|f(u) - f(S_i)\|^2$$
$$= q_1 \sum_{u,v \in C_1, u \neq v} \|f(u) - f(v)\|^2 + q_2 \sum_{u,v \in C_2, u \neq v} \|f(u) - f(v)\|^2$$
$$+ q_{12} \sum_{u \in C_1, v \in C_2} \|f(u) - f(v)\|^2.$$

*Proof.* Let $I_u$ be the index such that $u \in S_{I_u}$. We have

$$\mathbb{E}_{P \sim \mathcal{P}} \sum_{S_i \in S} \sum_{u \in S_i} s \|f(u) - f(S_i)\|^2 = \mathbb{E}_{P \sim \mathcal{P}} \sum_{S_i \in S} \frac{1}{2} \sum_{u \in S_i} \sum_{v \in S_i} \|f(u) - f(v)\|^2 \tag{17}$$

For fixed $S_i \in S$, without loss of generality we assume $f(S_i) = 0$ (if not, redefine $f(u) = f(u) - f(S_i)$), then we have

$$\frac{1}{2} \sum_{u \in S_i} \sum_{v \in S_i} \|f(u) - f(v)\|^2 \tag{18}$$

$$= \frac{1}{2} \sum_{u \in S_i} \sum_{v \in S_i} (\|f(u)\|^2 + \|f(v)\|^2 - 2f(u)^T f(v)) \tag{19}$$

$$= \sum_{u \in S_i} s \|f(u) - f(S_i)\|^2. \tag{20}$$

The equation 20 is due to the assumption that $f(S_i) = 0$. Therefore,

$$\mathbb{E}_{P \sim \mathcal{P}} \sum_{S_i \in S} \sum_{u \in S_i} s \|f(u) - f(S_i)\|^2 = \mathbb{E}_{P \sim \mathcal{P}} \sum_{u \neq v} \mathbb{I}_{[I_u = I_v]} \|f(u) - f(v)\|^2. \tag{21}$$

We next divide pairs in (21) into three categories and get

$$
\mathbb{E}_{P\sim\mathcal{P}}\sum_{u\neq v}\mathbb{I}_{[I_u=I_v]}\|f(u)-f(v)\|^2
$$
$$
=\mathbb{E}_{P\sim\mathcal{P}}\sum_{u,v\in C_1,u\neq v}\mathbb{I}_{[I_u=I_v]}\|f(u)-f(v)\|^2
$$
$$
+\mathbb{E}_{P\sim\mathcal{P}}\sum_{u,v\in C_2,u\neq v}\mathbb{I}_{[I_u=I_v]}\|f(u)-f(v)\|^2
$$
$$
+\mathbb{E}_{P\sim\mathcal{P}}\sum_{u\in C_1,v\in C_2}\mathbb{I}_{[I_u=I_v]}\|f(u)-f(v)\|^2
$$
$$
=\sum_{u,v\in C_1,u\neq v}q_1\|f(u)-f(v)\|^2+\sum_{u,v\in C_2,u\neq v}q_2\|f(u)-f(v)\|^2
$$
$$
+\sum_{u\in C_1,v\in C_2}q_{12}\|f(u)-f(v)\|^2,
$$

which finishes the proof. $\qquad\square$

### A.2 Comparison of Node Embeddings of the Original Graph and the Coarsened Graph

In this section, we assume the graph is connected. When there are multiple connected components, each component can be analyzed separately, and thus the conclusion holds for general graphs.

The feature matrix of the coarsened graph is computed using the formula in line 13 of Algorithm 1, which is different from prior work. Here we provide a justification on this. We consider a GNN encoder $Z = \sigma(\hat{A}^k XW)$ with $\hat{A} = \widetilde{D}^{-1}\widetilde{A}$. Assume the corresponding supernode of $u$ in the coarsened graph is $v$. We use $Z_u$ and $Z'_v$ to represent the node embeddings learned from the original graph $G$ and the coarsened graph $G'$ respectively. We show next that using our coarsened feature matrix, the difference between $Z_u$ and $Z'_v$ converges to zero as $k \to \infty$.

We assume the activation function $\sigma(\cdot)$ and the linear transformation function $W$ to be Lipschitz continuous. These assumptions are commonly used in previous analyses of GNNs (Chen et al., 2018; Garg et al., 2020; Cong et al., 2020; 2021). Then, the coarsening error can be expressed as:

$$
\|Z_u - Z'_v\| = \|\sigma(\hat{A}^k XW)_u - \sigma(\hat{A}'^k X'W)_v\| \le \kappa\|(\hat{A}^k X)_u - (\hat{A}'^k X')_v\|, \tag{22}
$$

where $\kappa$ represents the Lipschitz constant. For notational convenience, let $\pi_u^{(k)} = (\hat{A}^k X)_u$ and $\pi_v'^{(k)} = (\hat{A}'^k X')_v$. We need the following Lemma, the proof of which can be found in Chung & Graham (1997).

**Lemma A.2.**
$$
\hat{A}_{i,j}^\infty = \frac{\widetilde{d}_j}{\sum_{u\in G}\widetilde{d}_u} = \frac{\widetilde{d}_j}{2m+n}, \qquad |\hat{A}_{i,j}^k - \hat{A}_{i,j}^\infty| \le \lambda_2^k \widetilde{d}_i^{-\frac{1}{2}}\widetilde{d}_j^{\frac{1}{2}}, \tag{23}
$$

*where $\lambda_2$ is the second largest eigenvalue of $\hat{A}$ and $\widetilde{d}_i$ denotes the degree of node $i$ with self-loop.*

**Theorem A.3.** *Let the coarsened feature $X' = \widetilde{D}'^{-1}P^T\widetilde{D}X$, then for any node $u$, we have*

$$
\|\pi_u^{(k)} - \pi_v'^{(k)}\| \le \sqrt{\frac{d_{max}}{d_{min}}}(n\lambda_2^k + n'\lambda_2'^k), \tag{24}
$$

*where $\lambda_2$ and $\lambda_2'$ are the second largest eigenvalues in $G$ and $G'$ respectively. $d_{max}$ and $d_{min}$ are the maximum degree and minimum degree in $G$ and $G'$, respectively.*

*Proof.* Given the coarsened adjacency matrix $\widetilde{A}' = P^T\widetilde{A}P$, the sum of the weighted edges in $\widetilde{A}'$ is still $2m + n$. If node $j$ in $G'$ is not a supernode, we have $\hat{A}'^\infty_{i,j} = \frac{\widetilde{d'_j}}{\sum_{v\in G'}\widetilde{d'_v}} = \frac{\widetilde{d_j}}{2m+n} = \hat{A}_{i,j}^\infty$. Let $S$ be the set of

supernodes in $G'$, i.e., those nodes in $G'$ containing at least two nodes from the original graph, and define $Q$ as the set of nodes participating in the coarsening process: $Q = \bigcup_{S_i \in S} S_i$. Then we have:

$$
\begin{aligned}
&\|\pi_u^{(k)} - \pi_v'^{(k)}\| \\
=&\| \sum_{S_i \in S} (\hat{A}'^k_{v,S_i} \cdot X'_{S_i} - \sum_{j \in S_i} \hat{A}^k_{u,j} \cdot X_j) + \sum_{j \in V \setminus Q} (\hat{A}^k_{u,j} \cdot X_j - \hat{A}'^k_{v,j} \cdot X'_j)\| \\
\leq&\| \sum_{S_i \in S} (\hat{A}'^k_{v,S_i} \cdot X'_{S_i} - \sum_{j \in S_i} \hat{A}^k_{u,j} \cdot X_j))\| + \| \sum_{j \in V \setminus Q} (\hat{A}^k_{u,j} \cdot X_j - \hat{A}'^k_{v,j} \cdot X'_j)\|.
\end{aligned}
\tag{25}
$$

First,

$$
\begin{aligned}
&\| \sum_{S_i \in S} (\hat{A}'^k_{v,S_i} \cdot X'_{S_i} - \sum_{j \in S_i} \hat{A}^k_{u,j} \cdot X_j)\| \\
=&\| \sum_{S_i \in S} (\hat{A}'^k_{v,S_i} \cdot X'_{S_i} - \hat{A}'^\infty_{v,S_i} \cdot X'_{S_i} + \hat{A}'^\infty_{v,S_i} \cdot X'_{S_i} \\
&- \sum_{j \in S_i} (\hat{A}^k_{u,j} \cdot X_j - \hat{A}^\infty_{u,j} \cdot X_j + \hat{A}^\infty_{u,j} \cdot X_j))\| \\
=&\| \sum_{S_i \in S} (\hat{A}'^k_{v,S_i} \cdot X'_{S_i} - \hat{A}'^\infty_{v,S_i} \cdot X'_{S_i} - \sum_{j \in S_i} (\hat{A}^k_{u,j} \cdot X_j - \hat{A}^\infty_{u,j} \cdot X_j) \\
&+ \hat{A}'^\infty_{v,S_i} \cdot X'_{S_i} - \sum_{j \in S_i} \hat{A}^\infty_{u,j} \cdot X_j)\| \\
\leq&\| \sum_{S_i \in S} (\hat{A}'^k_{v,S_i} \cdot X'_{S_i} - \hat{A}'^\infty_{v,S_i} \cdot X'_{S_i})\| + \| \sum_{S_i \in S} \sum_{j \in S_i} (\hat{A}^k_{u,j} \cdot X_j - \hat{A}^\infty_{u,j} \cdot X_j)\| \\
&+ \| \sum_{S_i \in S} (\hat{A}'^\infty_{v,S_i} \cdot X'_{S_i} - \sum_{j \in S_i} \hat{A}^\infty_{u,j} \cdot X_j)\| \\
\leq& \sum_{S_i \in S} \|\hat{A}'^k_{v,S_i} - \hat{A}'^\infty_{v,S_i}\| \|X'_{S_i}\| + \sum_{S_i \in S} \sum_{j \in S_i} \|\hat{A}^k_{u,j} - \hat{A}^\infty_{u,j}\| \|X_j\| \\
&+ \| \sum_{S_i \in S} (\hat{A}'^\infty_{v,S_i} \cdot X'_{S_i} - \sum_{j \in S_i} \hat{A}^\infty_{u,j} \cdot X_j)\|.
\end{aligned}
\tag{26}
$$

For the sake of simplicity, we assume the feature $X_j$ is non-negative and normalize the $X_j$ so that $\|X_j\| = 1$. Let the supernode feature $X'_{S_i} = \frac{\sum_{j \in S_i} \hat{A}^\infty_{u,j} X_j}{\hat{A}'^\infty_{v,S_i}} = \frac{\sum_{j \in S_i} \widetilde{d_j} X_j}{\sum_{j \in S_i} \widetilde{d_j}}$. In other words, the coarsened feature matrix $X'$ is defined as $\widetilde{D}'^{-1} P^T \widetilde{D} X$, which implies $\|X'_j\| \leq 1$. Then, we have

$$
\begin{aligned}
&\| \sum_{S_i \in S} (\hat{A}'^k_{v,S_i} \cdot X'_{S_i} - \sum_{j \in S_i} \hat{A}^k_{u,j} \cdot X_j)\| \\
\leq& \sum_{S_i \in S} \|\hat{A}'^k_{v,S_i} - \hat{A}'^\infty_{v,S_i}\| \|X'_{S_i}\| + \sum_{S_i \in S} \sum_{j \in S_i} \|\hat{A}^k_{u,j} - \hat{A}^\infty_{u,j}\| \|X_j\| \\
&+ \| \sum_{S_i \in S} (\hat{A}'^\infty_{v,S_i} \cdot X'_{S_i} - \sum_{j \in S_i} \hat{A}^\infty_{u,j} \cdot X_j)\| \\
\leq& \sum_{S_i \in S} \|\hat{A}'^k_{v,S_i} - \hat{A}'^\infty_{v,S_i}\| + \sum_{S_i \in S} \sum_{j \in S_i} \|\hat{A}^k_{u,j} - \hat{A}^\infty_{u,j}\| \\
\leq& \sum_{S_i \in S} \lambda_2'^k \widetilde{d_u}^{-\frac{1}{2}} \widetilde{d_{S_i}}^{\frac{1}{2}} + \sum_{S_i \in S} \sum_{j \in S_i} \lambda_2^k \widetilde{d_u}^{-\frac{1}{2}} \widetilde{d_j}^{\frac{1}{2}}. \quad \text{(by Lemma A.2)}
\end{aligned}
\tag{27}
$$

For each uncoarsened node $u \in V \setminus Q$, we have $\hat{A}^\infty_{u,j} = \hat{A}'^\infty_{v,j}$ and $X_j = X'_j$. Therefore,

$$\| \sum_{j \in V \backslash Q} (\hat{A}_{u,j}^k \cdot X_j - \hat{A'}_{v,j}^k \cdot X'_j) \|$$

$$= \| \sum_{j \in V \backslash Q} (\hat{A}_{u,j}^k \cdot X_j - \hat{A}_{u,j}^\infty \cdot X_j + \hat{A}_{u,j}^\infty \cdot X_j - \hat{A'}_{v,j}^k \cdot X'_j) \|$$

$$\leq \sum_{j \in V \backslash Q} (\| \hat{A}_{u,j}^k \cdot X_j - \hat{A}_{u,j}^\infty \cdot X_j \| + \| \hat{A}_{u,j}^\infty \cdot X_j - \hat{A'}_{v,j}^k \cdot X'_j \|) \quad (28)$$

$$\leq \sum_{j \in V \backslash Q} (\| \hat{A}_{u,j}^k - \hat{A}_{u,j}^\infty \| + \| \hat{A'}_{v,j}^\infty - \hat{A'}_{v,j}^k \|)$$

$$\leq \sum_{j \in V \backslash Q} (\lambda_2^k \widetilde{d_u}^{-\frac{1}{2}} \widetilde{d_j}^{\frac{1}{2}} + \lambda_2'^k \widetilde{d_v'}^{-\frac{1}{2}} \widetilde{d_j'}^{\frac{1}{2}}). \quad \text{(by Lemma A.2)}$$

We define $d_{max}$ and $d_{min}$ as the maximum degree and minimum degree in $G$ and $G'$, respectively. Thus, we have the following conclusion

$$\| \pi_u^{(k)} - \pi_v'^{(k)} \|$$

$$\leq \sqrt{\frac{d_{max}}{d_{min}}} |S| \lambda_2'^k + \sqrt{\frac{d_{max}}{d_{min}}} |Q| \lambda_2^k + \sqrt{\frac{d_{max}}{d_{min}}} (n - |Q|)(\lambda_2^k + \lambda_2'^k)$$

$$= \sqrt{\frac{d_{max}}{d_{min}}} (n \lambda_2^k + (n + |S| - |Q|) \lambda_2'^k) \quad (29)$$

$$= \sqrt{\frac{d_{max}}{d_{min}}} (n \lambda_2^k + n' \lambda_2'^k).$$

$\square$

According to Theorem A.3 and inequality (22), we have the following upper bound:

$$\| Z_u - Z_v' \| \leq \kappa \sqrt{\frac{d_{max}}{d_{min}}} (n \lambda_2^k + n' \lambda_2'^k). \quad (30)$$

On the account of $0 < \lambda_2 < 1$ and $0 < \lambda_2' < 1$, when $k \to \infty$, the error $\| Z_u - Z_v' \| \to 0$.

## A.3 Generalizability of RGCCL

Following (Huang et al., 2023; Wang et al., 2022), we investigate the generalizability of our self-supervised model. Denote $\mathcal{G}_{v_i}$ as the ego network of node $v_i$ and the corresponding adjacency matrix as $\hat{A}_{\mathcal{G}_{v_i}}$, Wang et al. (2022) characterizes the concentration of a graph augmentation set with the following definition.

**Definition 2** $((\alpha, \gamma, \hat{d})$-Augmentation). *The data augmentation set $A$, which includes the original graph, is a $(\alpha, \gamma, \hat{d})$-augmentation, if for each class $C_k$, there exists a main part $C_k^0 \subseteq C_k$ (i.e., $\mathbb{P}[v \in C_k^0] \geq \sigma \mathbb{P}[v \in C_k]$), where $\sup_{v_1, v_2 \in C_k^0} d_A(v_1, v_2) \leq \gamma(\frac{\mathcal{D}}{\hat{d}_{\min}^k})^{1/2}$ hold with $d_A(v_i, v_j) = \min_{\mathcal{G}_i' \in A(\mathcal{G}_{v_i}), \mathcal{G}_j' \in A(\mathcal{G}_{v_j})} \| (\frac{\hat{A}_{\mathcal{G}_i'}}{\hat{d}_{\mathcal{G}_i'}} - \frac{\hat{A}_{\mathcal{G}_j'}}{\hat{d}_{\mathcal{G}_j'}}) X \|$, and $\hat{d}_{\min}^k$ is the minimum degree in class $C_k$.*

Given a $(\alpha, \gamma, \hat{d})$-Augmentation, one can establish inequalities between contrastive losses and the dot product of class center pairs. This means classes will be more separable while the objectives are being optimized. For example, Huang et al. (2023) gives the proofs for InfoNCE and the cross-correlation loss. Since our objective differs from both of them, we prove a similar theorem as Huang et al. (2023). Denote $\mu_k := \mathbb{E}_{v \in C_k} \mathbb{E}_{\mathcal{G}' \in A(\mathcal{G}_v)}[f(\mathcal{G}')]$, $p_k := \mathbb{P}[v \in C_k]$, $S_\epsilon := \{v \in \bigcup_{k=1}^K C_k : \forall \mathcal{G}_1, \mathcal{G}_2 \in A(\mathcal{G}_v), \| f(\mathcal{G}_1) - f(\mathcal{G}_2) \| \leq \epsilon\}$ and $R_\epsilon := 1 - \mathbb{P}[S_\epsilon]$.

**Theorem A.4.** *Assume that encoder $f$ with norm 1 is $M$-Lipschitz continuous. For a given $(\alpha, \gamma, \hat{d})$ augmentation set $A$, any $\epsilon > 0$ and $k \neq l$,*

$$\mu_k^T \mu_l \leq \frac{1}{p_k p_l}(-\frac{1}{2n^2 \mathcal{L}_{neg}} + \tau(\epsilon, \alpha, \gamma, \hat{d})), \tag{31}$$

*where* $\tau(\epsilon, \alpha, \gamma, \hat{d}) = 2R_\epsilon + 16(1 - \alpha(1 - \frac{1}{2}\epsilon - \frac{1}{4}M\max_k(\gamma\sqrt{\frac{\mathcal{D}}{\hat{d}_{\min}^k}})) + KR_\epsilon)^2 + 8(1 - \alpha(1 - \frac{1}{2}\epsilon - \frac{1}{4}M\max_k(\gamma\sqrt{\frac{\mathcal{D}}{\hat{d}_{\min}^k}})) + KR_\epsilon + \frac{K-1}{K})$.

*Proof.* Our negative pair loss is

$$\mathcal{L}_{neg} = \mathbb{E}_{P \sim \mathcal{P}}[\frac{1}{\sum_{i=1}^n \sum_{j=1}^n [\|h_i\|^2 + \|h_j\|^2 - 2h_i^T h_j]}]$$

$$= \frac{1}{2}\mathbb{E}_{P \sim \mathcal{P}}[\frac{1}{n^2 - \sum_{i=1}^n \sum_{j=1}^n h_i^T h_j}].$$

Since $\sum_{i=1}^n \sum_{j=1}^n h_i^T h_j \leq n^2$, and $\frac{1}{n^2 - x}$ is convex for $x \leq n^2$. Using Jensen's inequality, we have:

$$2\mathcal{L}_{neg} \geq \frac{1}{n^2 - \mathbb{E}_{P \sim \mathcal{P}}[\sum_{i=1}^n \sum_{j=1}^n h_i^T h_j]}$$

$$= \frac{1}{n^2(1 - \mathbb{E}_{x_i, x_j} \mathbb{E}_{P \sim \mathcal{P}}[h_i^T h_j])}.$$

Next, we focus on $\mathbb{E}_{x_i, x_j} \mathbb{E}_{P \sim \mathcal{P}} h_i^T h_j$:

$$\mathbb{E}_{x_i, x_j} \mathbb{E}_{P \sim \mathcal{P}}[h_i^T h_j] \geq \sum_{k=1}^K \sum_{l=1}^K \mathbb{E}_{x_i, x_j} \left[ \mathbb{I}(x_i \in S_\epsilon \cap C_k)\mathbb{I}(x_j \in C_l)\mathbb{E}_{P \sim \mathcal{P}}(h_i^T h_j) \right]$$

$$- \mathbb{E}_{x_i, x_j}[\mathbb{I}(x_i \in \bar{S}_\epsilon)]$$

$$= \sum_{k=1}^K \sum_{l=1}^K \mathbb{E}_{x_i, x_j} \left[ \mathbb{I}(x_i \in C_k)\mathbb{I}(x_j \in C_l)(\mu_k^T \mu_l) \right] - R_\epsilon + \Delta_1$$

$$= \sum_{k=1}^K \sum_{l=1}^K \left[ p_k p_l \mu_k^T \mu_l \right] - R_\epsilon + \Delta_1$$

$$\geq p_k p_l \mu_k^T \mu_l + \frac{1}{K} - R_\epsilon + \Delta_1,$$

where $\Delta_1$ is defined as

$$\Delta_1 = \sum_{k=1}^K \sum_{l=1}^K \mathbb{E}_{x_i, x_j}[\mathbb{I}(x_i \in S_\epsilon \cap C_k)\mathbb{I}(x_j \in C_l)\mathbb{E}_{P \sim \mathcal{P}}(h_i^T h_j)]$$

$$- \sum_{k=1}^K \sum_{l=1}^K \mathbb{E}_{x_i, x_j}[\mathbb{I}(x_i \in C_k)\mathbb{I}(x_j \in C_l)(\mu_k^T \mu_l)]$$

$$= \sum_{k=1}^K \sum_{l=1}^K \mathbb{E}_{x_i, x_j}[\mathbb{I}(x_i \in C_k)\mathbb{I}(x_j \in C_l)\mathbb{E}_{P \sim \mathcal{P}}[h_i^T h_j - \mu_k^T \mu_l]]$$

$$- \sum_{k=1}^K \sum_{l=1}^K \mathbb{E}_{x_i, x_j}[(\mathbb{I}(x_i \in C_k) - \mathbb{I}(x_i \in S_\epsilon \cap C_k))\mathbb{I}(x_j \in C_l)\mathbb{E}_{P \sim \mathcal{P}}(h_i^T h_j)].$$

Then,

$$
|\Delta_1| \leq R_\epsilon + \sum_{k=1}^{K}\sum_{l=1}^{K} \mathbb{E}_{x_i,x_j}\left[\mathbb{I}(x_i \in C_k)\mathbb{I}(x_j \in C_l)\mathbb{E}_{P \sim \mathcal{P}}|h_i^T h_j - \mu_k^T \mu_l|\right]
$$

$$
\leq R_\epsilon + 16(1 - \alpha(1 - \frac{1}{2}\epsilon - \frac{1}{4}M\max_k(\gamma\sqrt{\frac{\mathcal{D}}{\hat{d}_{\min}^k}})) + KR_\epsilon)^2
$$

$$
+ 8(1 - \alpha(1 - \frac{1}{2}\epsilon - \frac{1}{4}M\max_k(\gamma\sqrt{\frac{\mathcal{D}}{\hat{d}_{\min}^k}})) + KR_\epsilon).
$$

Thus, if we define

$$
\tau(\epsilon,\alpha,\gamma,\hat{d}) = 2R_\epsilon + 16(1 - \alpha(1 - \frac{1}{2}\epsilon - \frac{1}{4}M\max_k(\gamma\sqrt{\frac{\mathcal{D}}{\hat{d}_{\min}^k}})) + KR_\epsilon)^2
$$

$$
+ 8(1 - \alpha(1 - \frac{1}{2}\epsilon - \frac{1}{4}M\max_k(\gamma\sqrt{\frac{\mathcal{D}}{\hat{d}_{\min}^k}})) + KR_\epsilon + \frac{K-1}{K}),
$$

we have

$$
\mu_k^T \mu_l \leq \frac{1}{p_k p_l}(1 - \frac{1}{2n^2 \mathcal{L}_{neg}} - \frac{1}{K} + R_\epsilon + |\Delta_1|)
$$

$$
\leq \frac{1}{p_k p_l}(-\frac{1}{2n^2 \mathcal{L}_{neg}} + \tau(\epsilon,\alpha,\gamma,\hat{d})).
$$

This finishes the proof. □

### A.4 Random Graph Coarsening Algorithm

Algorithm 1 is a detailed description of our random graph coarsening algorithm.

---
**Algorithm 1** Random Graph Coarsening
---
**Input:** $G = (A, X)$, threshold $\delta$, the coarsening ratio $r$, number of nodes $n$
**Output:** $G' = (A', X')$
  Compute the weight set $\mathcal{I}$ of all edges
  Construct an edge set $\mathcal{E}$ of length $rn$ by randomly selecting the edges according to the weight set $\mathcal{I}$.
  Initialize the cluster list $\mathcal{T}$
  **for** $i = 0$ to $rn$ **do**
    Obtain $(u, v) = \mathcal{E}_i$
    Retrieve the clusters $\mathcal{T}_u$ and $\mathcal{T}_v$ from $\mathcal{T}$, where $\mathcal{T}_u$ contains $u$ and $\mathcal{T}_v$ contains $v$
    **if** $|\mathcal{T}_u| + |\mathcal{T}_v| < \delta$ and $\mathcal{T}_u \neq \mathcal{T}_v$ **then**
      Merge cluster $\mathcal{T}_u$ and cluster $\mathcal{T}_v$ to a new cluster
    **end if**
  **end for**
  Construct the assignment matrix $P$ by $\mathcal{T}$
  Compute the coarsened adjacency matrix $A' = P^T A P$
  Compute the coarsened feature matrix $X' = \widetilde{D}'^{-1}P^T\widetilde{D}X$
  **Return** $G' = (A', X')$
---

### A.5 Experimental details

For all unsupervised models, the learned representations are evaluated by training and testing a logistic regression classifier except for Ogbn-Arxiv. Since Ogbn-arxiv exhibits more complex characteristics, we use a more powerful MLP classifier. The detailed statistics of the six datasets are summarized in Table 7.

Table 7: Summary of the datasets used in our experiments

| Dataset | Nodes | Features | Classes | Avg. Degree |
|---|---|---|---|---|
| Cora | 2,708 | 1,433 | 7 | 3.907 |
| Citeseer | 3,327 | 3,703 | 6 | 2.74 |
| Pubmed | 19,717 | 500 | 3 | 4.50 |
| Amazon-Photo | 7,650 | 745 | 8 | 31.13 |
| Amazon-Computers | 13,752 | 767 | 10 | 35.76 |
| Ogbn-Arxiv | 169,343 | 128 | 40 | 13.67 |

**Details of our model.** In our model, we use SGC as the encoder for Cora, Citeseer, Pubmed, while we use GCN as the encoder for Photo and Computers. The detailed hyperparameter settings are listed in Table 8. Our code is available at: `https://github.com/szzhang17/Understanding-Class-Bias-Amplification-in-Graph-Representation-Learning`.

Table 8: Summary of the hyper-parameters.

| Dataset | Epoch | Learning rate | $\alpha$ | $\beta$ |
|---|---|---|---|---|
| Cora | 25 | 0.01 | 15000 | 500 |
| Citeseer | 200 | 0.0002 | 15000 | 500 |
| Pubmed | 25 | 0.02 | 20000 | 200 |
| Amazon-Photo | 20 | 0.001 | 100000 | 100000 |
| Amazon-Computers | 20 | 0.0002 | 20000 | 20000 |
| Ogbn-Arxiv | 10 | 0.0001 | 2000000 | 200000 |

**Details of Baselines.** We compare RGCCL with state-of-the-art GCL models DGI[1], GRACE[2], GraphCL[3], GCA[4], CCA-SSG[5], gCooL[6], GGD[7], GRADE[8], SPAN[9], PolyGCL[10] and classic graph embedding model Deepwalk. For all the baseline models, we use the source code from corresponding repositories. Due to the large scale of Ogbn-Arixv, some GCL models are unable to process the full-graph on GPU because of memory limitations. As a result, we apply graph sampling techniques to train these models.

**Configuration.** All the algorithms and models are implemented in Python and PyTorch Geometric. Experiments are conducted on a server with NVIDIA 3090 GPU (24 GB memory), NVIDIA A6000 GPU (48 GB memory) and Intel(R) Xeon(R) Silver 4210R CPU @ 2.40GHz.

---

[1]DGI (MIT License): `https://github.com/pyg-team/pytorch_geometric/blob/master/examples/infomax_transductive.py`

[2]GRACE (Apache License 2.0): `https://github.com/CRIPAC-DIG/GRACE`

[3]GraphCL (MIT License): `https://github.com/Shen-Lab/GraphCL`

[4]GCA (MIT License): `https://github.com/CRIPAC-DIG/GCA`

[5]CCA-SSG (Apache License 2.0): `https://github.com/hengruizhang98/CCA-SSG`

[6]gCooL (MIT License): `https://github.com/lblaoke/gCooL`

[7]GGD (MIT License): `https://github.com/zyzisastudyreallyhardguy/Graph-Group-Discrimination`

[8]GRADE (MIT License): `https://github.com/BUPT-GAMMA/Uncovering-the-Structural-Fairness-in-Graph-Contrastive-Learning`

[9]SPAN (MIT License): `https://github.com/Louise-LuLin/GCL-SPAN`

[10]PolyGCL (MIT License): `https://github.com/ChenJY-Count/PolyGCL`

