# OpenReview forum: "Understanding Class Bias Amplification in Graph Representation Learning"
_TMLR — Accepted by TMLR_

### Review · Reviewer_kEHx · 2025-09-22

**Summary Of Contributions:**

**Summary:**

This paper explores a structural bias phenomenon in graph representation learning (GRL), referred to as class bias amplification. It is the tendency of graph neural network (GNN) encoders to worsen performance disparities across classes. The authors present a theoretical analysis to explain how such bias arises in downstream classification tasks. And, building on these findings, they introduce a method based on contrastive learning to alleviate the issue.

**Strengths:**

1.	The paper is well-motivated, with a clear progression from problem definition to theoretical analysis and method design.
2.	The paper establishes a sufficient theoretical foundation to explain the phenomenon of bias amplification.
3.	Extensive experiments on multiple benchmark datasets validate the effectiveness of the proposed method.

**Weaknesses:**

1.	The most recent baseline used for comparison is from 2023. It would be helpful to present empirical comparisons with newer methods.
2.	The ablation study and sensitivity analyses (e.g., hyperparameters, coarsening strategies) are relatively brief and would benefit from deeper exploration.
3.	Since the main focus of the paper is addressing class bias amplification, the related work section should include a discussion of prior studies on bias amplification and clarify how the proposed notion of class bias amplification differs from existing definitions of bias amplification [1,2,3].

[1] Zhao, Jieyu, et al. "Men also like shopping: Reducing gender bias amplification using corpus-level constraints." arXiv preprint arXiv:1707.09457 (2017).

[2] Wang, Angelina, and Olga Russakovsky. "Directional bias amplification." International Conference on Machine Learning. PMLR, 2021.

[3] Hall, Melissa, et al. "A systematic study of bias amplification." arXiv preprint arXiv:2201.11706 (2022).

**Audience:**

Yes

**Audience Explanation:**

The paper addresses an important problem in graph representation learning—the tendency of GNN encoders to amplify performance disparities across classes—a challenge that is highly relevant to both the graph learning and bias mitigation research communities.

**Claims And Evidence:**

Yes

**Claims Explanation:**

The paper’s claims are supported with both theoretical and empirical evidence. The theoretical analysis offers a clear explanation of class bias amplification, while the proposed random graph coarsening and RGCCL model are well-motivated. Experiments on six benchmark datasets consistently demonstrate reduced class disparities and improved overall accuracy.

**Requested Changes:**

1.	The flow of the paper is disrupted by Section 5; I suggest moving it to the end, just before the conclusion.
2.	The comparison with recent fairness-aware GNN methods is limited and should be expanded.
3.	The ablation study should be presented in greater detail.
4.	The related work section should include a discussion on bias amplification and clarify how the proposed notion of class bias amplification differs from prior definitions.

---

> ### Author Response · Authors · 2025-10-16
>
> Dear Reviewer kEHx, ﻿
>
> We sincerely thank you for your kind and constructive feedback on our manuscript. Below is our response.
>
> > W1: "The most recent baseline used for comparison is from 2023. It would be helpful to present empirical comparisons with newer methods."
>
> Thank you for your valuable suggestion. We have added a new baseline, PolyGCL (ICLR 2024). Our proposed RGCCL still achieves the best performance.
>
> > W2 & R3: "The ablation study and sensitivity analyses (e.g., hyperparameters, coarsening strategies) are relatively brief and would benefit from deeper exploration." “The ablation study should be presented in greater detail.”
>
> Following your suggestion, we have added an ablation study on the loss function, which can be found in Table 6 of the revised version. This study aims to quantify the contribution of each component by separately removing the positive sample term ($L_{pos}$) and the negative sample term ($L_{neg}$). The results strongly validate the rationality of our loss function design.
>
> > W3 & R4: "Since the main focus of the paper is addressing class bias amplification, the related work section should include a discussion of prior studies on bias amplification and clarify how the proposed notion of class bias amplification differs from existing definitions of bias amplification [1,2,3]."
>
> Thank you for your suggestion. We have added relevant discussions in the related work section.
>
> > R1: The flow of the paper is disrupted by Section 5; I suggest moving it to the end, just before the conclusion.
>
> We have followed your suggestion and moved the related work section to before the conclusion.
>
> > R2：The comparison with recent fairness-aware GNN methods is limited and should be expanded.
>
> We thank the reviewer for raising the point about comparisons with fairness-aware GNNs. Our work focuses specifically on *unsupervised* graph representation learning for *structural* bias, which is a distinct and less explored problem. As we now clarify in the revised manuscript, most existing fairness-aware GNNs are designed for *supervised* learning and address *feature-associated* bias, making a direct comparison under our unsupervised setting methodologically misaligned. The method GRADE was included as it is one of the few unsupervised GCL works that explicitly targets a structural imbalance.

---

### Review · Reviewer_3oVj · 2025-09-30

**Summary Of Contributions:**

The paper identifies class bias amplification in GNN-based graph representation learning: when GNN encoders are used for unsupervised pretraining, downstream per-class performance gaps can increase relative to non-graph baselines. The authors formalize how local structural disparities between classes induce different message-passing convergence speeds, causing embedding concentration discrepancies and biased class-wise errors. They propose RGCCL (Random Graph Coarsening Contrastive Learning), using stochastic graph coarsening that favors low-degree regions. The method  creates a two-view contrastive objective: each node is pulled toward its coarsened super-node embedding, with negatives sampled only from the coarsened view. Experiments on six node-classification benchmarks show improved Macro-F1 and reduced per-class disparity versus strong GCL baselines. Ablations section examines memory use and coarsening ratios.

Strengths
- The authors define a clear articulation of a practically relevant phenomenon (class-level performance gaps in GRL) with aspectral explanation (different second eigenvalues and different finite-depth convergence)
- They theoretically analyze and propose a simple yet effective strategy (general augmentation, i.e. random coarsening ) backed by analysis (q₂>q₁>q₁₂) and integrated into a contrastive framework
- Broad, competitive empirical study across standard datasets with different models and ablations studies on embeddings properties define a well designed experimental setup

Weaknesses
- While the manuscript includes links to baseline code, it does not (as far as I can see) provide a code repository for RGCCL, which would be important for reproducibility
- The related work section is too brief and would benefit from a more comprehensive discussion and positioning within the broader literature, for example by expanding it incorporating section A.5 from the appendix
- The paper does not explicitly discuss its limitations, for example how the proposed method behaves on heterophilic versus homophilic graphs

**Additional Comments:**

Some typos:
	- Sec.1 Introduction: "behaviour:They try"
	- Sec 3.:"to the same class.C_1"

**Audience:**

Yes

**Audience Explanation:**

Yes. Graph contrastive learning and fairness in graph representation learning are active topics, and the idea of using coarsening as a bias reducing augmentation is simple, practical, and broadly applicable

**Broader Impact Concerns:**

I recommend adding a brief clarification that the notion of fairness in this work is limited to class-wise performance balance in benchmark tasks and does not extend to demographic or sensitive groups.

**Claims And Evidence:**

Yes

**Claims Explanation:**

Mostly yes. The paper provides a spectral account of local convergence leading to embedding concentration imbalance, a statistical link from variance imbalance to class error asymmetry, and experiments on multiple datasets showing overall gains and reduced dispersion across classes compared to strong baselines. This supports the main acceptance criterion of TMLR

**Requested Changes:**

- The paper does not explicitly discuss its limitations. In particular, all experiments are conducted on standard homophilic benchmarks (Cora, Citeseer, PubMed, Amazon, OGB-Arxiv), while it remains unclear how the proposed method would perform on heterophilic graphs (e.g., Chameleon, Texas). Since the theoretical analysis is built on assumptions of community homogeneity, some discussion of this limitation and potential extensions to heterophilous settings would strengthen the work.
- Provide a public code repository for RGCCL to ensure reproducibility, in addition to the baseline code links.
- Expand the related work section with a more comprehensive discussion and clearer positioning within the broader literature, for example by integrating or extending section A.5 from the appendix.

---

> ### Author Response · Authors · 2025-10-16
>
> Dear Reviewer 3oVj, ﻿
>
> Thank you for your thorough and insightful review. Below is our response.
>
>
>
> > W1 & RC2:"While the manuscript includes links to baseline code, it does not (as far as I can see) provide a code repository for RGCCL, which would be important for reproducibility." "Provide a public code repository for RGCCL to ensure reproducibility, in addition to the baseline code links."
>
> Thank you for your suggestion. We have provided the source code in the supplementary material, and the code links for all the baselines are included in the appendix.
>
>
>
> > W2 & RC3: "The related work section is too brief and would benefit from a more comprehensive discussion and positioning within the broader literature, for example by expanding it incorporating section A.5 from the appendix." "Expand the related work section with a more comprehensive discussion and clearer positioning within the broader literature, for example by integrating or extending section A.5 from the appendix."
>
> Thank you for your suggestion. We have integrated the related work from the appendix into the main text and further expanded the discussion on the references related to bias amplification.
>
> > W3 & RC1: "The paper does not explicitly discuss its limitations, for example how the proposed method behaves on heterophilic versus homophilic graphs""The paper does not explicitly discuss its limitations. In particular, all experiments are conducted on standard homophilic benchmarks (Cora, Citeseer, PubMed, Amazon, OGB-Arxiv), while it remains unclear how the proposed method would perform on heterophilic graphs (e.g., Chameleon, Texas). Since the theoretical analysis is built on assumptions of community homogeneity, some discussion of this limitation and potential extensions to heterophilous settings would strengthen the work."
>
> We sincerely thank the reviewer for this valuable feedback and for pointing out the limitation regarding heterophilic graphs. We acknowledge that our paper would be strengthened by a explicit discussion of this scope. The current work focuses on homophilic graphs to build a coherent theory under the community homogeneity assumption, and we recognize that this is a limitation in terms of generalizability.
>
> The question of how structural and class biases operate in heterophilic settings is, in our view, a profound one that warrants dedicated investigation. We believe it is not merely an extension but a separate research thrust, given the divergent graph properties. A discussion of this limitation has been added to the Section 7, where we also outline our future plans to investigate bias in heterophily, potentially through new metrics and theoretical adaptations. We are grateful for the insight that inspired this direction.
>
> > About typos:
>
> Thank you for your suggestion. We have corrected the relevant typos.

---

### Review · Reviewer_dkPn · 2025-10-01

**Summary Of Contributions:**

This paper deals with the problem of bias favoring well-connected nodes (that tend to correlate with some classes, possibly majority classes) in Graph Representation Learning (GRL) methods. It alternates nicely between intuition on simple cases and more general theory.

After explaining the issue and demonstrating it very clearly on a concrete case (figure 1), it proposes an explanation for it as mechanism that is both explained intuitively and backed by more theoretical derivations.
It then introduces a simple graph coarsening mechanism, RGCCL, that should alleviate the issue. It then demonstrates empirically how RGCCL effectively decreases the problem.

# Strengths:

- clearly written, nice to read, intuitions precede the maths
- a complete approach, as the paper combines theory (with intuitions), from problem observed, to theoretical solution, to experiments.


# Weaknesses

- it seems authors were stressed about the page limit and squashed a few things to appendix when they could have been integrated in the main text
- although overall clear, some definitions are missing. See requested changes.
- some details about the recipe for reproducing experiments (on new datasets) are missing. See requested changes.

**Additional Comments:**

I invite the authors to consider participating in:
- Unifying Perspectives on Learning Biases
- EurIPS 2025 Workshop - Copenhagen, DK - December 2025
- https://sites.google.com/view/towards-a-unified-view

**Audience:**

Yes

**Audience Explanation:**

This paper is at the frontier of several topics of general interest:
- Graph Representation Learning (Self Supervised)
- Bias and fairness in ML
- class imbalance

**Broader Impact Concerns:**

None, on the contrary, dealing with biases better can only help.

**Claims And Evidence:**

Yes

**Claims Explanation:**

The theory is backed by proofs presented in appendix,
but most importantly the intuitions presented are convincing (up to a few precisions required below)
and the experiments are rigorous and show the point claimed.

**Requested Changes:**

# Important changes
- remember that the length limit in TMLR is not a hard threshold, it's just that reviewing is supposed to be done earlier for papers less than 14 pages. Don't squash too much relevant material into the appendix, it hurts clarity.
- please define mathematically the notions of 1) embedding density (or embedding concentration) and 2) class density (the sentence "C1 is denser than C2" above Eq 10 is also problematic). The only sentence I found was "Firstly, we measure the concentration for each class by calculating their mean distances of the embedding from the centroid [inline equation]"... at page 10. I think this should come earlier and with a numbered equation.  Although these notions are quite well explained intuitively, it is important to know what you mean quantitatively. I believe that actually estimating these densities is computationally hard, but one may look at the variance of the points (representations) within one class, or one cluster, which seems to be what is done.
- The reasoning after Eq. 10 is a bit fast for me.
    - 1. "homophily principle": there are counter examples to that principle I guess, it would be better to mention it. Although I guess than when classes mix together, there is less class bias amplification (at least by the mechanism discussed in the paper)
    - 2. * we adopted a simple probability function ...". This is a bit vague and could be stated more explicitly. This goes together with my next comment:
- lines 3,4 of algorithm 1 ("Compute the weight...to the weight set I.") which are the key lines, should be explicited, in the main text.  This is a central part of the RGCCL recipe, so it should be clearly stated and explained in the main text. The rest is rather technical and straightforward and can be kept in appendix (well the coarsened feature matrix computation is non trivial but is discussed in the main text, so, fine.)
- paragraph just before section 5, "Next we discuss the concentration property of the random coarsening augmentation." is quite unclear. I follow well the beginning, but not how it relates to the conclusion: "Once u and v are coarsened together in at least one coarsened graph, d(u, v) = 0, which means our random coarsening augmentation can be very well-concentrated." How come? I don't see the connection at all with the fact that any two nodes, however far in the graph, can always be connected, at least with probability $p^{diameter(G)}$.
- how were α,β chosen ? In table 7, values are given, but how are they guessed? It's important to provide guidelines for future users.
- is code available? Please make it so.


# Small changes and Typos
- the statement "However, class bias amplification occurs in GNN-based GRL and is unrelated to label distributions." seems slightly misleading. I think I understand the point, yet, the measures used later in the paper refer to unequal performance for different classes. Although GRL does not use labels to be trained, the problem of a weak representation for minority classes is surely related to class frequency, isn't it?
- typo page 3, missing space: belonging to the same class.C 1 is more
- typo page 6, missing space: thus omitted.The above
- table 3 does not follow the style of other tables, also the order of the other models has been changed. Please always order them the same way.
- table 5: you could add a comment about how results are very robust to changes in $r$! It's actually quite impressive. Often, people introduce new methods, that actually work well only when fine-tuning their newly introduced hyper-parameters (here, r), That's not the case at all here, very nice.
- "A.5 More Related Work" -> move to main text.
- algorithm 1: define n.

---

> ### Author Response · Authors · 2025-10-16
>
> Dear Reviewer dkPn, ﻿
>
> We sincerely thank you for your kind and constructive feedback on our manuscript. Below is our response.
>
>
> > remember that the length limit in TMLR is not a hard threshold...
>
> Thank you for your suggestion. We have integrated the related work from the appendix into the main text and further expanded the discussion on the references related to bias amplification.
>
>
>
>
>
> > please define mathematically the notions ... which seems to be what is done.
>
>
>
> The embedding density in Section 3.1 is defined by $d_\mathcal{M}(H^{(L)})$, and the specific concept of $d_\mathcal{M}(H^{(L)})$ can be found in Section 2. Since $d_\mathcal{M}(H^{(L)})$ is difficult to compute, we use $V_C = \frac{1}{|C|}\sum_{i \in C} |z_i - \frac{1}{|C|} \sum_{i \in C} z_i|$ as a substitute in the experimental section.
>
>
>
>
>
> > "homophily principle": there are counter examples to that principle I guess...
>
> The problem discussed in this paper is based on homophilic graphs, so heterophilic graphs can be considered as counter examples. A detailed discussion on this point is provided in the newly added Section 7.
>
>
>
>
>
> > lines 3,4 of algorithm 1 ("Compute the weight...to the weight set I.") which are the key lines...
>
> Thank you for your suggestion. We have added the corresponding explanation after Equation (10).
>
>
>
>
>
> > paragraph just before section 5, "Next we discuss the concentration property of the random coarsening augmentation." is quite unclear...
>
>
>
>  In the coarsened graph, if two nodes are coarsened together, they will share the same node embedding in the coarsened graph. This means that during this training iteration, these two nodes obtain identical representations from the coarsening perspective, which can be viewed as d(u, v) = 0.
>
>
>
> > how were α,β chosen ? In table 7, values are given, but how are they guessed? It's important to provide guidelines for future users.
>
> We adopt the following strategy for the settings: β is typically set to be 5 to 20 times the number of nodes, while α is usually set to be close to the number of nodes. Additionally, we have included an ablation study in the experimental section to analyze the impact of the positive and negative sample terms in the loss function on the model's performance. This result can be found in Table 6 of the revised version. With the above settings, both positive and negative samples can contribute effectively.
>
>
>
>
>
> > is code available? Please make it so.
>
> Thank you for your suggestion. We have provided the source code in the supplementary material.
>
>
>
> > the statement "However, class bias amplification occurs in GNN-based GRL and is unrelated to label distributions." seems slightly misleading...
>
> Thank you for this insightful observation. We acknowledge that the original statement may have been prone to misinterpretation, as label imbalance may also lead to another form of bias during prediction. We have revised this statement to avoid such misunderstanding.
>
> > small typos.
>
> Thank you for your suggestion. We have corrected the relevant typos.

---

> > ### Comment · Reviewer_dkPn · 2025-10-27
> > **put a little bit more effort into changing manuscript to make it excellent**
> >
> > I thanks the authors for their detailed answers, and for the (few) changes to the manuscript. I think a bit more comments (of mine or other referees) could be included to further improve the manuscript, at a negligible cost for the authors. Overall, I do like this paper, so I think it's worth putting a tiny bit more effort into making it very clear to read.
> >
> > Overall, addressing the comments made by referees more thoroughly in the paper would help to further improve it.
> >
> > Here are some quick comments/answers:
> >
> > 1. I believe that embdding density, or rather how one can empirically estimate it, should be defined around the first time the concept is used, i.e. right after:
> > > Therefore, the embedding density of each community is primarily determined by nodes
> > > that constitute the community.
> >
> > in page 4, not page 10, like it is now.
> >
> >
> > 2. You answered the useful explanation:
> > > We adopt the following strategy for the settings: β is typically set to be 5 to 20 times the number of nodes, while α is usually set to be close to the number of nodes. Additionally, we have included an ablation study in the experimental section to analyze the impact of the positive and negative sample terms in the loss function on the model's performance. This result can be found in Table 6 of the revised version. With the above settings, both positive and negative samples can contribute effectively.
> >
> > But you did not include it in the paper. You really should.
> >
> > 3. Please also include the answer you made me, that I believe is useful for readers:
> > > In the coarsened graph, if two nodes are coarsened together, they will share the same node embedding in the coarsened graph. This means that during this training iteration, these two nodes obtain identical representations from the coarsening perspective, which can be viewed as d(u, v) = 0.
> >
> >
> >
> > 4. code availability
> >
> > > > is code available? Please make it so.
> > >
> > > Thank you for your suggestion. We have provided the source code in the > supplementary material.
> >
> > I see the zip, it's fine. I would put it on github, if I were you. And, crucially, refer to the code location, wherever it is, in the text of the paper.

---

### Decision · Action_Editor_sgTe · 2025-10-29

**Recommendation:** Accept as is

**Audience:**

Yes

**Audience Explanation:**

The paper is of interest to the graph representation learning community, which is highly represented in the TMLR audience. It provides a wide contribution that ranges from proposing a new potential issue, analyzing it theoretically, and proposing and validating a solution.

**Claims And Evidence:**

Yes

**Claims Explanation:**

The paper analyzes a phenomenon in GNNs they call "class bias amplification", wherein differences in performance across classes are amplified by the message passing operation (for homophilic graphs). They provide a theoretical analysis based on the local spectra induced by the class clusters. Based on this, they also propose a novel self-supervised algorithm based on random coarsening of the original graph.

All reviewers agree that the paper is addressing an interesting problem; that the theoretical analysis is valid, readable, and theoretically sound; and that the proposed method is well-supported by the previous theoretical analysis. In addition, the experimental evaluation is comprehensive (with one exception, see below). Limitations are clearly described in the final manuscript, especially w.r.t. heterophilic graphs.

During review, the reviewers highlighted some minor concerns, including the lack of recent (2024) baselines, variations in the flow of the paper, lack of ablations in the main text, and lack of source code for reproducibility. All these points were addressed during the review round, and all reviewers agree in accepting the paper.